# A Stochastic Optimization Framework for Fair Risk Minimization

**Andrew Lowy**[*]
University of Southern California
lowya@usc.edu

**Sina Baharlouei**[*]
University of Southern California
baharlou@usc.edu

**Rakesh Pavan**
Univerity of Washington
rakeshpavan333@gmail.com

**Meisam Razaviyayn**
University of Southern California
razaviya@usc.edu

**Ahmad Beirami**[†]
Google Research
beirami@google.com

## Abstract

Despite the success of large-scale empirical risk minimization (ERM) at achieving high accuracy across a variety of machine learning tasks, *fair* ERM is hindered by the incompatibility of fairness constraints with stochastic optimization. We consider the problem of fair classification with discrete sensitive attributes and potentially large models and data sets, requiring stochastic solvers. Existing in-processing fairness algorithms are either impractical in the large-scale setting because they require large batches of data at each iteration or they are not guaranteed to converge. In this paper, we develop *the first stochastic in-processing fairness algorithm with guaranteed convergence*. For demographic parity, equalized odds, and equal opportunity notions of fairness, we provide slight variations of our algorithm–called FERMI–and prove that each of these variations converges in stochastic optimization with any batch size. Empirically, we show that FERMI is amenable to stochastic solvers with multiple (non-binary) sensitive attributes and non-binary targets, performing well even with minibatch size as small as one. Extensive experiments show that FERMI achieves the most favorable tradeoffs between fairness violation and test accuracy across all tested setups compared with state-of-the-art baselines for demographic parity, equalized odds, equal opportunity. These benefits are especially significant with small batch sizes and for non-binary classification with large number of sensitive attributes, making FERMI a practical, scalable fairness algorithm.

## 1 Introduction

Ensuring that decisions made using machine learning (ML) algorithms are fair to different subgroups is of utmost importance. Without any mitigation strategy, learning algorithms may result in discrimination against certain subgroups based on sensitive attributes, such as gender or race, even if such discrimination is absent in the training data Mehrabi et al. [2021], and algorithmic fairness literature aims to remedy such discrimination issues Sweeney [2013], Datta et al. [2015], Feldman et al. [2015], Bolukbasi et al. [2016], Angwin et al. [2016], Calmon et al. [2017a], Hardt et al. [2016], Fish et al. [2016], Woodworth et al. [2017], Zafar et al. [2017], Bechavod and Ligett [2017], Agarwal et al. [2018], Kearns et al. [2018], Prost et al. [2019], Lahoti et al. [2020]. Modern ML problems often involve large-scale models with hundreds of millions or even billions of parameters, e.g., BART Lewis et al. [2019], ViT Dosovitskiy et al. [2020], GPT-2 Radford et al. [2019]. In such cases, during

---

[*]Equal contribution.
[†]Work done at Meta AI.

2022 Trustworthy and Socially Responsible Machine Learning (TSRML 2022) co-located with NeurIPS 2022.

| Reference | NB target | NB attrib. | NB code | Beyond logistic | Stoch. alg. (unbiased**) | Converg. (stoch.) |
|---|---|---|---|---|---|---|
| **FERMI (this work)** | ✓ | ✓ | ✓ | ✓ | ✓ (✓) | ✓ (✓) |
| Cho et al. [2020a] | ✓ | ✓ | ✓ | ✓ | ✓ (✗) | ✗ |
| Cho et al. [2020b] | ✓ | ✓ | ✗ | ✓ | ✓ (✓) | ✗ |
| Baharlouei et al. [2020] | ✓ | ✓ | ✓ | ✓ | ✗ | ✓ (✗) |
| Rezaei et al. [2020] | ✗ | ✗ | ✗ | ✗ | ✗ | ✗ |
| Jiang et al. [2020]* | ✗ | ✓ | ✗ | ✗ | ✗ | ✗ |
| Mary et al. [2019] | ✓ | ✓ | ✓ | ✓ | ✓ (✗) | ✗ |
| Prost et al. [2019] | ✗ | ✗ | ✗ | ✓ | ✓ (✗) | ✗ |
| Donini et al. [2018] | ✗ | ✓ | ✗ | ✓ | ✗ | ✗ |
| Zhang et al. [2018] | ✓ | ✓ | ✗ | ✓ | ✓ (✗) | ✗ |
| Agarwal et al. [2018] | ✗ | ✓ | ✗ | ✓ | ✗ | ✓ (✗) |

Table 1: Comparison of state-of-the-art in-processing methods (**NB = non-binary**) on whether they (a) handle non-binary targets (beyond binary classification), (b) handle non-binary sensitive attributes, (c) release code that applies to non-binary targets/attributes, (d) extend to arbitrary models, (e) provide code for stochastic optimization (and whether the gradients are unbiased), (f) provide convergence guarantees (for stochastic optimization). FERMI is the only method compatible with stochastic optimization and guaranteed convergence. The only existing baselines for non-binary classification with non-binary sensitive attributes are Mary et al. [2019], Baharlouei et al. [2020], Cho et al. [2020a] (NB code). *We refer to the in-processing method of Jiang et al. [2020], not their post-processing method. **We use the term "unbiased" in statistical estimation sense; not to be confused with bias in the fairness sense.

fine-tuning, the available memory on a node constrains us to use stochastic optimization with (small) minibatches in each training iteration. In this paper, we address the dual challenges of *fair* and *stochastic* ML, providing *the first stochastic fairness algorithm that provably converges with any batch size*.

A machine learning algorithm satisfies the *demographic parity* fairness notion if the predicted target is independent of the sensitive attributes Dwork et al. [2012]. Promoting demographic parity can lead to poor performance, especially if the true outcome is not independent of the sensitive attributes. To remedy this, Hardt et al. [2016] proposed *equalized odds* to ensure that the predicted target is conditionally independent of the sensitive attributes given the true label. A further relaxed version of this notion is *equal opportunity* which is satisfied if predicted target is conditionally independent of sensitive attributes given that the true label is in an advantaged class Hardt et al. [2016]. Equal opportunity ensures that false positive rates are equal across different demographics, where negative outcome is considered an advantaged class, e.g., extending a loan. See Appendix A for formal definitions of these fairness notions.

**Measuring fairness violation.** In practice, the learner only has access to finite samples and cannot verify demographic parity, equalized odds, or equal opportunity. This has led the machine learning community to define several fairness violation metrics that quantify the degree of (conditional) independence between random variables, e.g., $L_\infty$ distance Dwork et al. [2012], Hardt et al. [2016], mutual information Kamishima et al. [2011], Rezaei et al. [2020], Steinberg et al. [2020], Zhang et al. [2018], Cho et al. [2020b], Roh et al. [2020], Pearson correlation Zafar et al. [2017], Beutel et al. [2019], false positive/negative rate difference Bechavod and Ligett [2017], Hilbert Schmidt independence criterion (HSIC) Pérez-Suay et al. [2017], kernel-based minimum mean discrepancy (MMD) Prost et al. [2019], Rényi correlation Mary et al. [2019], Baharlouei et al. [2020], Grari et al. [2019, 2020], and exponential Rényi mutual information (ERMI) Mary et al. [2019]. In this paper, we focus on three variants of ERMI specialized to demographic parity, equalized odds, and equal opportunity. The motivation for the use of ERMI is two-fold. First, we will see in Sec. 2 that *ERMI is amenable to stochastic optimization*. Moreover, we observe (Appendix C) that ERMI provides an upper bound on several of the above notions of fairness violation. Consequently, a model trained to reduce ERMI will also provide guarantees on these other fairness violations.[3]

**Related work & contributions.** Fairness-promoting machine learning algorithms can be categorized in three main classes: *pre-processing*, *post-processing*, and *in-processing* methods. Pre-processing algorithms Feldman et al. [2015], Zemel et al. [2013], Calmon et al. [2017a] transform the biased data features to a new space in which the labels and sensitive attributes are statistically independent. This

---

[3]Nevertheless, we use $L_\infty$ distance for measuring fairness violation in our numerical experiments, since $L_\infty$ is broadly used.

transform is oblivious to the training procedure. Post-processing approaches Hardt et al. [2016], Pleiss et al. [2017] mitigate the discrimination of the classifier by altering the final decision. In-processing approaches focus on the training procedure and impose the notions of fairness as constraints or regularization terms in the training procedure. Several regularization-based methods are proposed in the literature to promote fairness Ristanoski et al. [2013], Quadrianto and Sharmanska [2017] in decision-trees Kamiran et al. [2010], Raff et al. [2018], Aghaei et al. [2019], support vector machines Donini et al. [2018], boosting Fish et al. [2015], neural networks Grari et al. [2020], Cho et al. [2020a], Prost et al. [2019], or (logistic) regression models Zafar et al. [2017], Berk et al. [2017], Taskesen et al. [2020], Chzhen and Schreuder [2020], Baharlouei et al. [2020], Jiang et al. [2020], Grari et al. [2019]. See the recent paper by Hort et al. [2022] for a more comprehensive literature survey.

While in-processing approaches generally give rise to better tradeoffs between fairness violation and performance, existing approaches are mostly incompatible with stochastic optimization. This paper addresses this problem in the context of fair (non-binary) classification with discrete (non-binary) sensitive attributes. See Table 1 for a summary of the main differences between FERMI and existing in-processing methods.

Our main contributions are as follows:

1. For each given fairness notion (demographic parity, equalized odds, or equal opportunity), we formulate an objective that uses ERMI as a regularizer to balance fairness and accuracy (Eq. (FRMI obj.)), and derive an empirical version of this objective (Eq. (FERMI obj.)). We propose an algorithm (Algorithm 1) for solving each of these objectives, which is *the first stochastic in-processing fairness algorithm with guaranteed convergence*. The main property needed to obtain a convergent stochastic algorithm is to derive a (stochastically) unbiased estimator of the gradient of the objective function. The existing stochastic fairness algorithms by Zhang et al. [2018], Mary et al. [2019], Prost et al. [2019], Cho et al. [2020b,a] are not guaranteed to converge since there is no straightforward way to obtain such unbiased estimator of the gradients for their fairness regularizers.[4] For any minibatch size (even as small as 1), we prove (Theorem 1) that our algorithm converges to an approximate solution of the empirical objective (Eq. (FERMI obj.)).

2. We show that if the number of training examples is large enough, then our algorithm (Algorithm 1) converges to an approximate solution of the *population*-level objective (Theorem 2). The proofs of these convergence theorems require the development of novel techniques (see e.g. Proposition 1 and Proposition 2), and the resourceful application of many classical results from optimization, probability theory, and statistics.

3. We demonstrate through extensive numerical experiments that our stochastic algorithm achieves superior fairness-accuracy tradeoff curves against all comparable baselines for demographic parity, equalized odds, and equal opportunity. In particular, *the performance gap is very large when minibatch size is small* (as is practically necessary for large-scale problems) and the number of sensitive attributes is large.

## 2   Fair Risk Minimization through ERMI Regularization

In this section, we propose a fair learning objective (Eq. (FRMI obj.)) and derive an empirical variation (Eq. (FERMI obj.)) of this objective. We then develop a stochastic optimization algorithm (Algorithm 1) that we use to solve these objectives, and prove that our algorithm converges to an approximate solution of the two objectives.

Consider a learner who trains a model to make a prediction, $\widehat{Y}$, e.g., whether or not to extend a loan, supported on $[m] := \{1, \ldots, m\}$. The prediction is made using a set of features, $\mathbf{X}$, e.g., financial history features. Assume that there is a set of discrete sensitive attributes, $S$, e.g., race and sex, supported on $[k]$.

We now define the fairness violation notion that we will use to enforce fairness in our model.

**Definition 1** (ERMI – exponential Rényi mutual information). *We define the exponential Rényi mutual information between random variables $\widehat{Y}$ and $S$ with joint distribution $p_{\widehat{Y}, S}$ and marginals*

---

[4]We suspect it might be possible to derive a provably convergent stochastic algorithm from the framework in Prost et al. [2019] using our techniques, but their approach is limited to binary classification with binary sensitive attributes. By contrast, we provide (empirical and population-level) convergence guarantees for our algorithm with any number of sensitive attributes and any number of classes.

$p_{\widehat{Y}}$, $p_S$ by:

$$D_R(\widehat{Y}; S) := \mathbb{E}\left\{\frac{p_{\widehat{Y},S}(\widehat{Y}, S)}{p_{\widehat{Y}}(\widehat{Y})p_S(S)}\right\} - 1 = \sum_{j\in[m]}\sum_{r\in[k]}\frac{p_{\widehat{Y},S}(j,r)^2}{p_{\widehat{Y}}(j)p_S(r)} - 1. \qquad \text{(ERMI)}$$

Definition 1 is what we would use if *demographic parity* were the fairness notion of interest. If instead one wanted to promote fairness with respect to equalized odds or equal opportunity, then it is straightforward to modify the definition by substituting appropriate conditional probabilities for $p_{\widehat{Y},S}$, $p_{\widehat{Y}}$, and $p_S$ in Eq. (ERMI): see Appendix B. In Appendix B, we also discuss that ERMI is the $\chi^2$-divergence (which is an $f$-divergence) between the joint distribution, $p_{\widehat{Y},S}$, and the Kronecker product of marginals, $p_{\widehat{Y}} \otimes p_S$ Calmon et al. [2017b]. In particular, ERMI is non-negative, and zero if and only if demographic parity (or equalized odds or equal opportunity, for the conditional version of ERMI) is satisfied. Additionally, we show in Appendix C that ERMI provides an upper bound on other commonly used measures of fairness violation: Shannon mutual information Cho et al. [2020b], Rényi correlation Baharlouei et al. [2020], $L_q$ fairness violation Kearns et al. [2018], Hardt et al. [2016]. Therefore, any algorithm that makes ERMI small will also have small fairness violation with respect to these other notions.

We can now define our fair risk minimization through exponential Rényi mutual information framework:[5]

$$\min_{\boldsymbol{\theta}}\left\{\text{FRMI}(\boldsymbol{\theta}) := \mathcal{L}(\boldsymbol{\theta}) + \lambda D_R\big(\widehat{Y}_{\boldsymbol{\theta}}(\mathbf{X}); S\big)\right\}, \qquad \text{(FRMI obj.)}$$

where $\mathcal{L}(\boldsymbol{\theta}) := \mathbb{E}_{(\mathbf{X},Y)}[\ell(\mathbf{X}, Y; \boldsymbol{\theta})]$ for a given loss function $\ell$ (e.g. $L_2$ loss or cross entropy loss); $\lambda > 0$ is a scalar balancing the accuracy versus fairness objectives; and $\widehat{Y}_{\boldsymbol{\theta}}(\mathbf{X})$ is the output of the learned model (i.e. the predicted label in a classification task). While $\widehat{Y}_{\boldsymbol{\theta}}(\mathbf{X}) = \widehat{Y}(\mathbf{X}; \boldsymbol{\theta})$ inherently depends on $\mathbf{X}$ and $\boldsymbol{\theta}$, in the rest of this paper, we sometimes leave the dependence of $\widehat{Y}$ on $\mathbf{X}$ and/or $\boldsymbol{\theta}$ implicit for brevity of notation. Notice that we have also left the dependence of the loss on the predicted outcome $\widehat{Y} = \widehat{Y}_{\boldsymbol{\theta}}(\mathbf{X})$ implicit.

As is standard, we assume that the prediction function satisfies $\mathbb{P}(\widehat{Y}(\boldsymbol{\theta}, \mathbf{X}) = j|\mathbf{X}) = \mathcal{F}_j(\boldsymbol{\theta}, \mathbf{X})$, where $\mathcal{F}(\boldsymbol{\theta}, \mathbf{X}) = (\mathcal{F}_1(\boldsymbol{\theta}, \mathbf{X}), \ldots, \mathcal{F}_m(\boldsymbol{\theta}, \mathbf{X}))^T \in [0,1]^m$ is differentiable in $\boldsymbol{\theta}$ and $\sum_{j=1}^m \mathcal{F}_j(\boldsymbol{\theta}, \mathbf{X}) = 1$. For example, $\mathcal{F}(\boldsymbol{\theta}, \mathbf{X})$ could represent the probability label given by a logistic regression model or the output of a neural network after softmax layer. Indeed, this assumption is natural for most classifiers. Further, even classifiers, such as SVM, that are not typically expressed using probabilities can often be well approximated by a classifier of the form $\mathbb{P}(\widehat{Y}(\boldsymbol{\theta}, \mathbf{X}) = j|\mathbf{X}) = \mathcal{F}_j(\boldsymbol{\theta}, \mathbf{X})$, e.g. by using Platt Scaling Platt et al. [1999], Niculescu-Mizil and Caruana [2005].

The work of Mary et al. [2019] considered the same objective Eq. (FRMI obj.), and tried to empirically solve it through a kernel approximation. We propose a different approach to solving this problem, which we shall describe below. Essentially, we express ERMI as a "max" function (Proposition 1), which enables us to re-formulate Eq. (FRMI obj.) (and its empirical counterpart Eq. (FERMI obj.)) as a stochastic min-max optimization problem. This allows us to use stochastic gradient descent ascent (SGDA) to solve Eq. (FRMI obj.). Unlike the algorithm of Mary et al. [2019], our algorithm provably *converges*. Our algorithm also empirically outperforms the algorithm of Mary et al. [2019], as we show in Sec. 3 and Appendix E.2.

## 2.1  A Convergent Stochastic Algorithm for Fair Empirical Risk Minimization

In practice, the true joint distribution of $(\mathbf{X}, S, Y, \widehat{Y})$ is unknown and we only have $N$ samples at our disposal. Let $\mathbf{D} = \{\mathbf{x}_i, s_i, y_i, \widehat{y}(\mathbf{x}_i; \boldsymbol{\theta})\}_{i\in[N]}$ denote the features, sensitive attributes, targets, and the predictions of the model parameterized by $\boldsymbol{\theta}$ for these given samples. For now, we consider the empirical risk minimization (ERM) problem and do not require any assumptions on the data set

---

[5]In this section, we present all results in the context of demographic parity, leaving off all conditional expectations for clarity of presentation. The algorithm/results are readily extended to equalized odds and equal opportunity by using the conditional version of Eq. (ERMI) (which is described in Appendix B); we use these resulting algorithms for numerical experiments.

(e.g. we allow for different samples in $\mathbf{D}$ to be drawn from different, heterogeneous distributions). Consider the empirical objective

$$\min_{\boldsymbol{\theta}} \left\{ \text{FERMI}(\boldsymbol{\theta}) := \widehat{\mathcal{L}}(\boldsymbol{\theta}) + \lambda \widehat{D}_R(\widehat{Y}_{\boldsymbol{\theta}}(\mathbf{X}), S) \right\}, \qquad \text{(FERMI obj.)}$$

where $\widehat{\mathcal{L}}(\boldsymbol{\theta}) := \frac{1}{N} \sum_{i=1}^{N} \ell(\mathbf{x}_i, y_i; \boldsymbol{\theta})$ is the empirical loss and[6]

$$\widehat{D}_R(\widehat{Y}, S) := \mathbb{E} \left\{ \frac{\hat{p}_{\widehat{Y}, S}(\widehat{Y}, S)}{\hat{p}_{\widehat{Y}}(\widehat{Y}) \hat{p}_S(S)} \right\} - 1 = \sum_{j \in [m]} \sum_{r \in [k]} \frac{\hat{p}_{\widehat{Y}, S}(j, r)^2}{\hat{p}_{\widehat{Y}}(j) \hat{p}_S(r)} - 1$$

is *empirical ERMI* with $\hat{p}$ denoting empirical probabilities with respect to $\mathbf{D}$: $\hat{p}_S(r) = \frac{1}{N} \sum_{i=1}^{N} \mathbb{1}_{\{s_i = r\}}$; $\hat{p}_{\hat{y}}(j) = \frac{1}{N} \sum_{i=1}^{N} \mathcal{F}_j(\boldsymbol{\theta}, \mathbf{x}_i)$; and $\hat{p}_{\widehat{Y}, S}(j, r) = \frac{1}{N} \sum_{i=1}^{N} \mathcal{F}_j(\boldsymbol{\theta}, \mathbf{x}_i) \mathbf{s}_{i, r}$ for $j \in [m], r \in [k]$. We shall see (Proposition 2) that empirical ERMI is a good approximation of ERMI when $N$ is large. Now, it is straightforward to derive an unbiased estimate for $\widehat{\mathcal{L}}(\boldsymbol{\theta})$ via $\frac{1}{|B|} \sum_{i \in B} \ell(\mathbf{x}_i, y_i; \boldsymbol{\theta})$ where $B \subseteq [N]$ is a random minibatch of data points drawn from $\mathbf{D}$. However, unbiasedly estimating $\widehat{D}_R(\widehat{Y}, S)$ in the objective function Eq. (FERMI obj.) with $|B| < N$ samples is more difficult. In what follows, we present our approach to deriving statistically *unbiased stochastic estimators* of the gradients of $\widehat{D}_R(\widehat{Y}, S)$ given a random batch of data points $B$. This stochastic estimator is key to developing a stochastic convergent algorithm for solving Eq. (FERMI obj.). The key novel observation that allows us to derive this estimator is that Equation FERMI obj. can be written as a *min-max* optimization problem (see Corollary 1). This observation, in turn, follows from the following result:

**Proposition 1.** *For random variables $\widehat{Y}$ and $S$ with joint distribution $\hat{p}_{\widehat{Y}, S}$, where $\widehat{Y} \in [m], S \in [k]$, we have*

$$\widehat{D}_R(\widehat{Y}; S) = \max_{W \in \mathbb{R}^{k \times m}} \left\{ -\operatorname{Tr}(W \widehat{P}_{\hat{y}} W^T) + 2 \operatorname{Tr}(W \widehat{P}_{\hat{y}, s} \widehat{P}_s^{-1/2}) - 1 \right\},$$

*if $\widehat{P}_{\hat{y}} = \operatorname{diag}(\hat{p}_{\widehat{Y}}(1), \dots, \hat{p}_{\widehat{Y}}(m))$, $\widehat{P}_s = \operatorname{diag}(\hat{p}_S(1), \dots, \hat{p}_S(k))$, and $(\widehat{P}_{\hat{y}, s})_{i, j} = \hat{p}_{\widehat{Y}, S}(i, j)$ with $\hat{p}_{\widehat{Y}}(i), \hat{p}_S(j) > 0$ for $i \in [m], j \in [k]$.*

The proof is a direct calculation, given in Appendix D. Let $\widehat{\mathbf{y}}(\mathbf{x}_i, \boldsymbol{\theta}) \in \{0, 1\}^m$ and $\mathbf{s}_i = (\mathbf{s}_{i,1}, \dots, \mathbf{s}_{i,k})^T \in \{0, 1\}^k$ be the one-hot encodings of $\widehat{y}(\mathbf{x}_i, \boldsymbol{\theta})$ and $s_i$, respectively for $i \in [N]$. Then, Proposition 1 provides a useful variational form of Eq. (FERMI obj.), which forms the backbone of our novel algorithmic approach:

**Corollary 1.** *Let $(\mathbf{x}_i, s_i, y_i)$ be a random draw from $\mathbf{D}$. Then, Eq. (FERMI obj.) is equivalent to*

$$\min_{\boldsymbol{\theta}} \max_{W \in \mathbb{R}^{k \times m}} \left\{ \widehat{F}(\boldsymbol{\theta}, W) := \widehat{\mathcal{L}}(\boldsymbol{\theta}) + \lambda \widehat{\Psi}(\boldsymbol{\theta}, W) \right\}, \qquad (1)$$

*where $\widehat{\Psi}(\boldsymbol{\theta}, W) = -\operatorname{Tr}(W \widehat{P}_{\hat{y}} W^T) + 2 \operatorname{Tr}(W \widehat{P}_{\hat{y}, s} \widehat{P}_s^{-1/2}) - 1 = \frac{1}{N} \sum_{i=1}^{N} \widehat{\psi}_i(\boldsymbol{\theta}, W)$ and*

$$\begin{aligned}
\widehat{\psi}_i(\boldsymbol{\theta}, W) &:= -\operatorname{Tr}(W \mathbb{E}[\widehat{\mathbf{y}}(\mathbf{x}_i, \boldsymbol{\theta}) \widehat{\mathbf{y}}(\mathbf{x}_i, \boldsymbol{\theta})^T | \mathbf{x}_i] W^T) + 2 \operatorname{Tr}(W \mathbb{E}[\widehat{\mathbf{y}}(\mathbf{x}_i; \boldsymbol{\theta}) \mathbf{s}_i^T | \mathbf{x}_i, s_i] \widehat{P}_s^{-1/2}) - 1 \\
&= -\operatorname{Tr}(W \operatorname{diag}(\mathcal{F}_1(\boldsymbol{\theta}, \mathbf{x}_i), \dots, \mathcal{F}_m(\boldsymbol{\theta}, \mathbf{x}_i)) W^T) + 2 \operatorname{Tr}(W \mathbb{E}[\widehat{\mathbf{y}}(\mathbf{x}_i; \boldsymbol{\theta}) \mathbf{s}_i^T | \mathbf{x}_i, s_i] \widehat{P}_s^{-1/2}) - 1.
\end{aligned}$$

Corollary 1 implies that for any given data set $\mathbf{D}$, the quantity $\ell(\mathbf{x}_i, y_i; \boldsymbol{\theta}) + \lambda \widehat{\psi}_i(\boldsymbol{\theta}, W)$ is an unbiased estimator of $\widehat{F}(\boldsymbol{\theta}, W)$ (with respect to the uniformly random draw of $i \in [N]$). Thus, we can use stochastic optimization (e.g. SGDA) to solve Eq. (FERMI obj.) with any batch size $1 \leq |B| \leq N$, and the resulting algorithm will be guaranteed to converge since the stochastic gradients are unbiased. We present our proposed algorithm, which we call *FERMI*, for solving Eq. (FERMI obj.) in Algorithm 1.

Note that the matrix $\widehat{P}_s^{-1/2}$ depends only on the full data set of sensitive attributes $\{s_1, \cdots, s_N\}$ and has no dependence on $\boldsymbol{\theta}$, and can therefore be computed just once, in line 2 of Algorithm 1. On the other hand, the quantities $\mathbb{E}[\widehat{\mathbf{y}}(\mathbf{x}_i, \boldsymbol{\theta}) \widehat{\mathbf{y}}(\mathbf{x}_i, \boldsymbol{\theta})^T | \mathbf{x}_i]$ and $\mathbb{E}[\widehat{\mathbf{y}}(\mathbf{x}_i; \boldsymbol{\theta}) \mathbf{s}_i^T | \mathbf{x}_i, s_i]$ depend on the sample $(\mathbf{x}_i, s_i, \widehat{y}_i)$ that is drawn in a given iteration and on the model parameters $\boldsymbol{\theta}$, and are therefore computed at each iteration of the algorithm.

---

[6]We overload notation slightly here and use $\mathbb{E}$ to denote expectation with respect to the *empirical* (joint) distribution.

---

**Algorithm 1** FERMI Algorithm

---

1: **Input**: $\boldsymbol{\theta}^0 \in \mathbb{R}^{d_\theta}$, $W^0 = 0 \in \mathbb{R}^{k \times m}$, step-sizes $(\eta_\theta, \eta_w)$, fairness parameter $\lambda \geq 0$, iteration number $T$, minibatch sizes $|B_t|, t \in \{0, 1, \cdots, T\}$, $\mathcal{W} :=$ Frobenius norm ball of radius $D$ around $0 \in \mathbb{R}^{k \times m}$ for $D$ given in Appendix D.
2: Compute $\widehat{P}_s^{-1/2} = \text{diag}(\hat{p}_S(1)^{-1/2}, \ldots, \hat{p}_S(k)^{-1/2})$.
3: **for** $t = 0, 1, \ldots, T$ **do**
4:      Draw a mini-batch $B_t$ of data points $\{(\mathbf{x}_i, s_i, y_i)\}_{i \in B_t}$
5:      Set $\boldsymbol{\theta}^{t+1} \leftarrow \boldsymbol{\theta}^t - \frac{\eta_\theta}{|B_t|} \sum_{i \in B_t} [\nabla_\theta \ell(\mathbf{x}_i, y_i; \boldsymbol{\theta}^t) + \lambda \nabla_\theta \widehat{\psi}_i(\boldsymbol{\theta}^t, W^t)]$.
6:      Set $W^{t+1} \leftarrow \Pi_{\mathcal{W}} \Big( W^t + \frac{2\lambda \eta_w}{|B_t|} \sum_{i \in B_t} \big[ -W^t \mathbb{E}[\widehat{\mathbf{y}}(\mathbf{x}_i, \boldsymbol{\theta}) \widehat{\mathbf{y}}(\mathbf{x}_i, \boldsymbol{\theta})^T | \mathbf{x}_i] + \widehat{P}_s^{-1/2} \mathbb{E}[\mathbf{s}_i \widehat{\mathbf{y}}^T(\mathbf{x}_i; \boldsymbol{\theta}^t) | \mathbf{x}_i, s_i] \big] \Big)$
7: **end for**
8: Pick $\hat{t}$ uniformly at random from $\{1, \ldots, T\}$.
9: **Return:** $\boldsymbol{\theta}^{\hat{t}}$.

---

Although the min-max problem Eq. (FERMI obj.) that we aim to solve is unconstrained, we project the iterates $W^t$ (in line 5 of Algorithm 1) onto a bounded set $\mathcal{W}$ in order to satisfy a technical assumption that is needed to prove convergence of Algorithm 1[7]. We choose $\mathcal{W}$ to be a sufficiently large ball that contains $W^*(\boldsymbol{\theta}) := \arg\max_W \widehat{F}(\boldsymbol{\theta}, W)$ for every $\boldsymbol{\theta}$ in some neighborhood of $\boldsymbol{\theta}^* \in \arg\min_{\boldsymbol{\theta}} \max_W \widehat{F}(\boldsymbol{\theta}, W)$, so that Eq. (FERMI obj.) is equivalent to

$$\min_{\boldsymbol{\theta}} \max_{W \in \mathcal{W}} \left\{ \widehat{F}(\boldsymbol{\theta}, W) = \widehat{\mathcal{L}}(\boldsymbol{\theta}) + \lambda \widehat{\Psi}(\boldsymbol{\theta}, W) \right\}.$$

See Appendix D for details. When applying Algorithm 1 in practice, it is not necessary to project the iterates; e.g. in Sec. 3, we obtain strong empirical results without projection in Algorithm 1.

Since Eq. (FERMI obj.) is potentially nonconvex in $\boldsymbol{\theta}$, a global minimum might not exist and even computing a local minimum is NP-hard in general Murty and Kabadi [1985]. Thus, as is standard in the nonconvex optimization literature, we aim for the milder goal of finding an approximate *stationary point* of Eq. (FERMI obj.). That is, given any $\epsilon > 0$, we aim to find a point $\boldsymbol{\theta}^*$ such that $\mathbb{E}\|\nabla \text{FERMI}(\boldsymbol{\theta}^*)\| \leq \epsilon$, where the expectation is solely with respect to the randomness of the algorithm (minibatch sampling). The following theorem guarantees that Algorithm 1 will find such a point efficiently:

**Theorem 1.** *(Informal statement) Let $\epsilon > 0$. Assume that $\ell(\mathbf{x}, y; \cdot)$ and $\mathcal{F}(\cdot, \mathbf{x})$ are Lipschitz continuous and differentiable with Lipschitz continuous gradient (see Appendix D for definitions), $\hat{p}_S(j) > 0$ for all sensitive attributes $j \in [k]$ and $\hat{p}_{\widehat{Y}}(l) \geq \mu > 0$ for all labels $l \in [m]$ and at every iterate $\boldsymbol{\theta}^t$. Then for any batch sizes $1 \leq |B_t| \leq N$, Algorithm 1 converges to an $\epsilon$-first order stationary point of the Eq. (FERMI obj.) objective in $\mathcal{O}\left(\frac{1}{\epsilon^5}\right)$ stochastic gradient evaluations.*

The formal statement of Theorem 1 can be found in Theorem 3 in Appendix D. Theorem 1 implies that Algorithm 1 can efficiently achieve any tradeoff between fairness (ERMI) violation and (empirical) accuracy, depending on the choice of $\lambda$.[8] However, if smaller fairness violation is desired (i.e. if larger $\lambda$ is chosen), then the algorithm needs to run for more iterations (see Appendix D). The proof of Theorem 1 follows from Corollary 1 and the observation that $\widehat{\psi}_i$ is strongly concave in $W$ (see Lemma 11 in Appendix D). This implies that Eq. (1) is a nonconvex-strongly concave min-max problem, so the convergence guarantee of SGDA Lin et al. [2020] yields Theorem 1.[9] The detailed proof of Theorem 1 is given in Appendix D. Increasing the batch size to $\Theta(\epsilon^{-2})$ improves the stochastic gradient complexity to $\mathcal{O}(\epsilon^{-4})$. On the other hand, increasing the batch size further to $|B_t| = N$ results in a deterministic algorithm which is guaranteed to find a point $\boldsymbol{\theta}^*$ such $\|\nabla \text{FERMI}(\boldsymbol{\theta}^*)\| \leq \epsilon$ (no expectation) in $\mathcal{O}(\epsilon^{-2})$ iterations [Lin et al., 2020, Theorem 4.4],[Ostrovskii et al., 2020, Remark 4.2]; this iteration complexity has the optimal dependence on $\epsilon$ Carmon et al. [2020], Zhang

---

[7]Namely, bounded $W^t$ ensures that the gradient of $\widehat{F}$ is Lipschitz continuous at every iterate and the variance of the stochastic gradients is bounded.

[8]This sentence is accurate to the degree that an approximate stationary point of the non-convex objective Eq. (FERMI obj.) corresponds to an approximate risk minimizer.

[9]A faster convergence rate of $\mathcal{O}(\epsilon^{-3})$ could be obtained by using the (more complicated) SREDA method of Luo et al. [2020] instead of SGDA to solve FERMI objective. We omit the details here.

et al. [2021]. However, like existing fairness algorithms in the literature, this full-batch variant is impractical for large-scale problems.

**Remark 1.** *The condition $\hat{p}_{\hat{Y}}(l) \geq \mu$ in Theorem 1 is assumed in order to ensure strong concavity of $\widehat{F}(\boldsymbol{\theta}^t, \cdot)$ at every iterate $\boldsymbol{\theta}^t$, which leads to the $\mathcal{O}(\epsilon^{-5})$ convergence rate. This assumption is typically satisfied in practice: for example, if the iterates $\boldsymbol{\theta}^t$ remain in a compact region during the algorithm and the classifier uses softmax, then $\hat{p}_{\hat{Y}}(l) \geq \mu > 0$. Having said that, it is worth noting that this condition is not absolutely necessary to ensure convergence of Algorithm 1. Even if this condition doesn't hold, then Eq. (1) is still a nonconvex-concave min-max problem. Hence SGDA still converges to an $\epsilon$-stationary point, albeit at the slower rate of $\mathcal{O}(\epsilon^{-8})$ Lin et al. [2020]. Alternatively, one can add a small $\ell_2$ regularization term to the objective to enforce strong concavity and get the fast convergence rate of $\mathcal{O}(\epsilon^{-5})$.*

## 2.2 Asymptotic Convergence of Algorithm 1 for Population-level FRMI Objective

So far, we have let $N \geq 1$ be arbitrary and have not made any assumptions on the underlying distribution(s) from which the data was drawn. Even so, we showed that Algorithm 1 always converges to a stationary point of Eq. (FERMI obj.). Now, we will show that if $\mathbf{D}$ contains *i.i.d.* samples from an unknown joint distribution $\mathcal{D}$ and if $N \gg 1$, then Algorithm 1 converges to an approximate solution of the *population* risk minimization problem Eq. (FRMI obj.). Precisely, we will use a one-pass sample-without-replacement ("online") variant of Algorithm 1 to obtain this population loss guarantee. The one-pass variant is identical to Algorithm 1 except that: a) once we draw a batch of samples $B_t$, we remove these samples from the data set so that they are never re-used; and b) the **for**-loop terminates when we have used all $n$ samples.

**Theorem 2.** *Let $\epsilon > 0$. Assume that $\ell(\mathbf{x}, y; \cdot)$ and $\mathcal{F}(\cdot, \mathbf{x})$ are Lipschitz continuous and differentiable with Lipschitz continuous gradient, and that $\min_{r \in [k]} p_S(r) > 0$. Then, there exists $N \in \mathbb{N}$ such that if $n \geq N$ and $\mathbf{D} \sim \mathcal{D}^n$, then a one-pass sample-without-replacement variant of Algorithm 1 converges to an $\epsilon$-first order stationary point of the Eq. (FRMI obj.) objective in $\mathcal{O}\left(\frac{1}{\epsilon^5}\right)$ stochastic gradient evaluations, for any batch sizes $|B_t|$.*

Theorem 2 provides a guarantee on the fairness/accuracy loss that can be achieved on unseen "test data." This is important because the main goal of (fair) machine learning is to (fairly) give accurate predictions on test data, rather than merely fitting the training data well. Specifically, Theorem 2 shows that with enough (i.i.d.) training examples at our disposal, (one-pass) Algorithm 1 finds an approximate stationary point of the population-level fairness objective Eq. (FRMI obj.). Furthermore, the gradient complexity is the same as it was in the empirical case. The proof of Theorem 2 will be aided by the following result, which shows that $\widehat{\psi}_i$ is an asymptotically unbiased estimator of $\Psi$, where $\max_W \Psi(\boldsymbol{\theta}, W)$ equals ERMI:

**Proposition 2.** *Let $\{z_i\}_{i=1}^n = \{\mathbf{x}_i, s_i, y_i\}_{i=1}^n$ be drawn i.i.d. from an unknown joint distribution $\mathcal{D}$. Denote $\widehat{\psi}_i^{(n)}(\boldsymbol{\theta}, W) = -\operatorname{Tr}(W\mathbb{E}[\widehat{\mathbf{y}}(\mathbf{x}_i, \boldsymbol{\theta})\widehat{\mathbf{y}}(\mathbf{x}_i, \boldsymbol{\theta})^T|\mathbf{x}_i]W^T) + 2\operatorname{Tr}\left(W\mathbb{E}[\widehat{\mathbf{y}}(\mathbf{x}_i; \boldsymbol{\theta})\mathbf{s}_i^T|\mathbf{x}_i, s_i]\left(\widehat{P}_s^{(n)}\right)^{-1/2}\right) - 1$, where $\widehat{P}_s^{(n)} = \frac{1}{n}\sum_{i=1}^n \operatorname{diag}(\mathbb{1}_{\{s_i=1\}}, \cdots, \mathbb{1}_{\{s_i=k\}})$. Denote $\Psi(\boldsymbol{\theta}, W) = -\operatorname{Tr}(WP_{\hat{y}}W^T) + 2\operatorname{Tr}(WP_{\hat{y},s}P_s^{-1/2}) - 1$, where $P_{\hat{y}} = \operatorname{diag}(\mathbb{E}\mathcal{F}_1(\boldsymbol{\theta}, \mathbf{x}), \cdots, \mathbb{E}\mathcal{F}_m(\boldsymbol{\theta}, \mathbf{x}))$, $(P_{\hat{y},s})_{j,r} = \mathbb{E}_{\mathbf{x}_i, s_i}[\mathcal{F}_j(\boldsymbol{\theta}, \mathbf{x}_i)\mathbf{s}_{i,r}]$ for $j \in [m], r \in [k]$, and $P_s = \operatorname{diag}(P_S(1), \cdots, P_S(k))$. Assume $p_S(r) > 0$ for all $r \in [k]$. Then,*

$$\max_W \Psi(\boldsymbol{\theta}, W) = D_R(\widehat{Y}(\boldsymbol{\theta}); S)$$

*and*

$$\lim_{n \to \infty} \mathbb{E}[\widehat{\psi}_i^{(n)}(\boldsymbol{\theta}, W)] = \Psi(\boldsymbol{\theta}, W).$$

The proof of Proposition 2 is given in Appendix D.1. The first claim is immediate from Proposition 1 and its proof, while the second claim is proved using the strong law of large numbers, the continuous mapping theorem, and Lebesgue's dominated convergence theorem.

Proposition 2 implies that the empirical stochastic gradients computed in Algorithm 1 are good approximations of the true gradients of Eq. (FRMI obj.). Intuitively, this suggests that when we use Algorithm 1 to solve the fair ERM problem Eq. (FERMI obj.), the output of Algorithm 1 will also be an approximate solution of Eq. (FRMI obj.). While Theorem 2 shows this intuition does indeed hold, the proof of Theorem 2 requires additional work. A reasonable first attempt at

proving Theorem 2 might be to try to bound the expected distance between the gradient of FRMI and the gradient of FERMI (evaluated at the point $\hat{\boldsymbol{\theta}}$ that is output by Algorithm 1) via Danskin's theorem Danskin [1966] and strong concavity, and then leverage Theorem 1 to conclude that the gradient of FRMI must also be small. However, the dependence of $\hat{\boldsymbol{\theta}}$ on the training data prevents us from obtaining a tight enough bound on the distance between the empirical and population gradients at $\hat{\boldsymbol{\theta}}$. Thus, we take a different approach to proving Theorem 2, in which we consider the output of two different algorithms: one is the conceptual algorithm that runs one-pass Algorithm 1 as if we had access to the true sensitive attributes $P_s$ ("Algorithm A"); the other is the realistic one-pass Algorithm 1 that only uses the training data ("Algorithm B"). We argue: 1) the output of the conceptual algorithm is a stationary point of the population-level objective; and 2) the distance between the gradients of the population-level objective at $\boldsymbol{\theta}_A$ and $\boldsymbol{\theta}_B$ is small. While 1) follows easily from the proof of Theorem 1 and the online-to-batch conversion, establishing 2) requires a careful argument. The main tools we use in the proof of Theorem 2 are Theorem 1, Proposition 2, Danskin's theorem, Lipschitz continuity of the $\arg\max$ function for strongly concave objective, the continuous mapping theorem, and Lebesgue's dominated convergence theorem: see Appendix D.1 for the detailed proof.

Note that the online-to-batch conversion used to prove Theorem 2 requires a *convergent stochastic optimization algorithm*; this implies that our arguments could not be used to prove an analogue of Theorem 2 for existing fair learning algorithms, since existing convergent fairness algorithms are not stochastic. An alternate approach to bounding the "generalization error" of our algorithm would be to use a standard covering/uniform convergence argument. However, this approach would not yield as tight a guarantee as Theorem 2. Specifically, the accuracy and/or gradient complexity guarantee would depend on the dimension of the space (i.e. the number of model parameters), since the covering number depends (exponentially) on the dimension. For large-scale problems with a huge number of model parameters, such dimension dependence is prohibitive.

As previously mentioned, we can interpret Theorem 2 as providing a guarantee that Algorithm 1 *generalizes* well, achieving small fairness violation and test error, even on unseen "test" examples–as long as the data is i.i.d. and $N$ is sufficiently large. In the next section, we empirically corroborate Theorem 2, by evaluating the fairness-accuracy tradeoffs of the FERMI algorithm (Algorithm 1) in several numerical experiments.

## 3 Numerical Experiments

In this section, we evaluate the performance of FERMI in terms of the fairness violation vs. test error for different notions of fairness (e.g. demographic parity, equalized odds, and equality of opportunity). To this end, we perform diverse experiments comparing FERMI to other state-of-the-art approaches on several benchmarks. In Section 3.1, we showcase the performance of FERMI applied to a logistic regression model on binary classification tasks with binary sensitive attributes on Adult, German Credit, and COMPAS datasets. In Section 3.2, we utilize FERMI with a convolutional neural network base model for fair (to different religious groups) toxic comment detection. In Section 3.3, we explore fairness in non-binary classification with non-binary sensitive attributes. Finally, Section 3.4 shows how FERMI may be used beyond fair empirical risk minimization in domain generalization problems to learn a model independent of spurious features.

### 3.1 Fair Binary Classification with Binary Sensitive Attributes using Logistic Regression

#### 3.1.1 Benchmarking full-batch performance

In the first set of experiments, we use FERMI to learn a fair logistic regression model on the Adult dataset. With the Adult data set, the task is to predict whether or not a person earns over \$50k annually without discriminating based on the sensitive attribute, gender. We compare FERMI against state-of-the-art in-processing full-batch ($|B| = N$) baselines, including Zafar et al. [2017], Feldman et al. [2015], Kamishima et al. [2011], Jiang et al. [2020], Hardt et al. [2016], Prost et al. [2019], Baharlouei et al. [2020], Rezaei et al. [2020], Donini et al. [2018], Cho et al. [2020a]. Since the majority of existing fair learning algorithms cannot be implemented with $|B| < N$, these experiments allow us to benchmark the performance of FERMI against a wider range of baselines. To contextualize the performance of these methods, we also include a **Naïve Baseline** that randomly replaces the model output with the majority label (0 in Adult dataset), with probability $p$ (independent of the data), and

sweep $p$ in $[0, 1]$. At one end ($p = 1$), the output will be provably fair with performance reaching that of a naive classifier that outputs the majority class. At the other end ($p = 0$), the algorithm has no fairness mitigation and obtains the best performance (accuracy). By sweeping $p$, we obtain a tradeoff curve between performance and fairness violation.

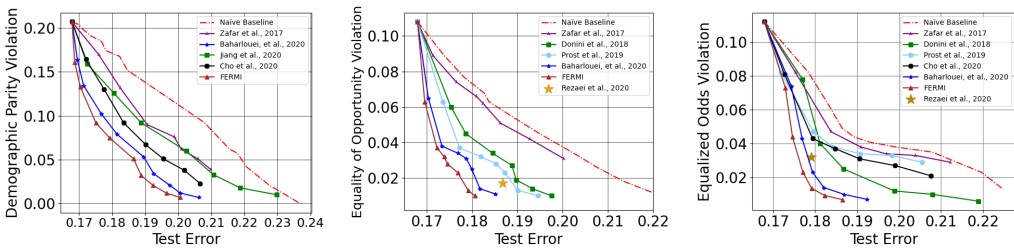

Figure 1: Accuracy/Fairness trade-off of FERMI and several state-of-the-art in-processing approaches on Adult dataset. FERMI offers the best fairness vs. accuracy tradeoff curve in all experiments against all baselines. Rezaei et al. [2020] only allow for a single output and do not yield a tradeoff curve. Further, the algorithms by Mary et al. [2019] and Baharlouei et al. [2020] are equivalent in this binary setting and shown by the red curve. In the binary/binary setting, FERMI, Mary et al. [2019] and Baharlouei et al. [2020] all try to solve the same objective Eq. (FRMI obj.). However, the empirical formulation Eq. (FERMI obj.) and FERMI algorithm that we use results in better performance, even though we are using a full-batch for all baselines in this experiment.

In Fig. 1, we report the fairness violation (demographic parity, equalized odds, and equality of opportunity violations) vs. test error of the aforementioned in-processing approaches on the Adult dataset. The upper left corner of the tradeoff curves coincides with the unmitigated baseline, which only optimizes for performance (smallest test error). As can be seen, FERMI offers a fairness-accuracy tradeoff curve that dominates all state-of-the-art baselines in each experiment and with respect to each notion of fairness, even in the full batch setting. Aside from in-processing approaches, we compare FERMI with several pre-processing and post-processing algorithms on Adult, German Credit, and COMPAS datasets in Appendix E.5, where we show that the tradeoff curves obtained from FERMI dominate that of all other baselines considered. See Appendix E for details on the data sets and experiments.

It is noteworthy that the empirical objectives of Mary et al. [2019] and Baharlouei et al. [2020] are exactly the same in the binary/binary setting, and their algorithms also coincide to the red curve in Fig. 1. This is because Exponential Rényi mutual information is equal to Rényi correlation for binary targets and/or binary sensitive attributes (see Lemma 2), which is the setting of all experiments in Sec. 3.1. Additionally, like us, in the binary/binary setting these works are trying to empirically solve Eq. (FRMI obj.), albeit using different estimation techniques; i.e., their empirical objective is different from Eq. (FERMI obj.). This demonstrates the effectiveness of our empirical formulation (FERMI obj.) and our solver (Algorithm 1), even though we are using all baselines in full batch mode in this experiment. See Appendix E.5 for the complete version of Fig. 1 which also includes pre-processing and post-processing baselines.

Fig. 8 in Appendix E illustrates that FERMI outperforms baselines in the presence of *noisy outliers* and *class imbalance*. Our theory did not consider the role of noisy outliers and class imbalance, so the theoretical investigation of this phenomenon could be an interesting direction for future work.

### 3.1.2 The effect of batch size on fairness/accuracy tradeoffs

Next, we evaluate the performance of FERMI on smaller batch sizes ranging from 1 to 64. To this end, we compare FERMI against several state-of-the-art in-processing algorithms that permit stochastic implementation for demographic parity: Mary et al. [2019], Baharlouei et al. [2020], and Cho et al. [2020a]. Similarly to the full batch setting, for all methods, we train a logistic regression model with a respective regularizer for each method. We use demographic parity $L_\infty$ violation (Definition 10) to measure demographic parity violation. More details about the dataset and experiments, and additional experimental results, can be found in Appendix E.

Fig. 2 shows that FERMI offers a superior fairness-accuracy tradeoff curve against all baselines, for each tested batch size, empirically confirming Theorem 1, as FERMI is the only algorithm that is guaranteed to converge for small minibatches. It is also noteworthy that all other baselines cannot

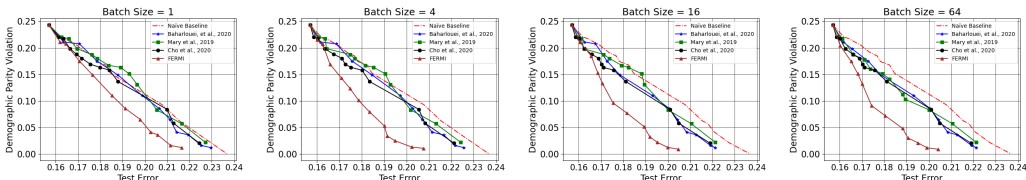

Figure 2: Performance of FERMI, Cho et al. [2020b], Mary et al. [2019], Baharlouei et al. [2020] with different batch-sizes on Adult dataset. FERMI demonstrates the best accuracy/fainess tradeoff across different batch sizes.

beat *Naïve Baseline* when the batch size is very small, e.g., $|B| = 1$. Furthermore, FERMI with $|B| = 4$ almost achieves the same fairness-accuracy tradeoff as the full batch variant.

### 3.1.3 The effect of missing sensitive attributes on fairness/accuracy tradeoffs

Sensitive attributes might be partially unavailable in many real-world applications due to legal issues, privacy concerns, and data gathering limitations Zhao et al. [2022], Coston et al. [2019]. Missing sensitive attributes make fair learning tasks more challenging in practice.

The unbiased nature of the estimator used in FERMI algorithm motivates that it may be able to handle cases where sensitive attributes are *partially* available and are dropped uniformly at random. As a case study on the Adult dataset, we randomly masked 90% of the sensitive attribute (i.e., gender entries). To estimate the fairness regularization term, we rely on the remaining 10% of the training samples ($\approx 3k$) with sensitive attribute information. Figure 3 depicts the tradeoff between accuracy and fairness (demographic parity) violation for FERMI and other baselines. We suspect that the superior accuracy-fairness tradeoff of FERMI compared to other approaches is due to the fact that the estimator of the gradient remains unbiased since the missing entries are missing completely at random (MCAR). Note that the *Naïve Baseline* is similar to the one implemented in the previous section, and *Full-sensitive FERMI* is an oracle method that applies FERMI to the data with no missing attributes (for comparison purposes only). We observe that FERMI achieves a slightly worse fairness-accuracy tradeoff compared to Full-sensitive FERMI oracle, whereas the other baselines are hurt significantly and are only narrowly outperforming the Naïve Baseline.

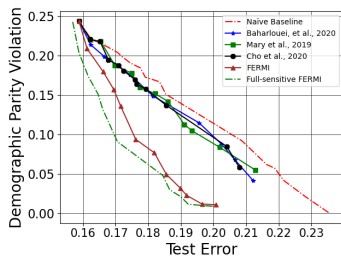

Figure 3: Performance of FERMI and other state-of-the-art approaches on the Adult dataset where 90% of gender entries are missing. Full-sensitive FERMI is obtained by applying FERMI on the data without any missing entries.

### 3.2 Fair Binary Classification using Neural Models

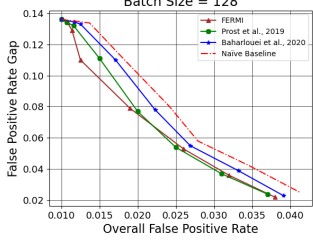
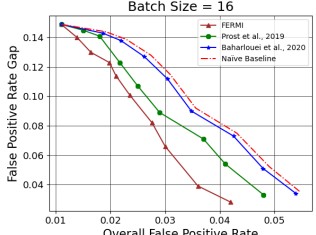

Figure 4: Fair toxic comment detection with different batch sizes. For $|B| = 128$, the performance of Prost et al. [2019] and FERMI are close to each other, however, when the batch size is reduced to 16, FERMI demonstrates a better fairness/ performance trade-off. The performance and fairness are measured by the overall false positive rate and the false positive gap between different religious sub-groups (Christians vs Muslim-Jews), respectively.

In this experiment, our goal is to showcase the efficacy of FERMI in stochastic optimization with neural network function approximation. To this end, we apply FERMI, Prost et al. [2019], Baharlouei et al. [2020], and Mary et al. [2019] (which coincides with Baharlouei et al. [2020]) to the Toxic Comment Classification dataset where the underlying task is to predict whether a given published comment in social media is toxic. The sensitive attribute is religion that is binarized into two groups: Christians in one group; Muslims and Jews in the other group. Training a neural network

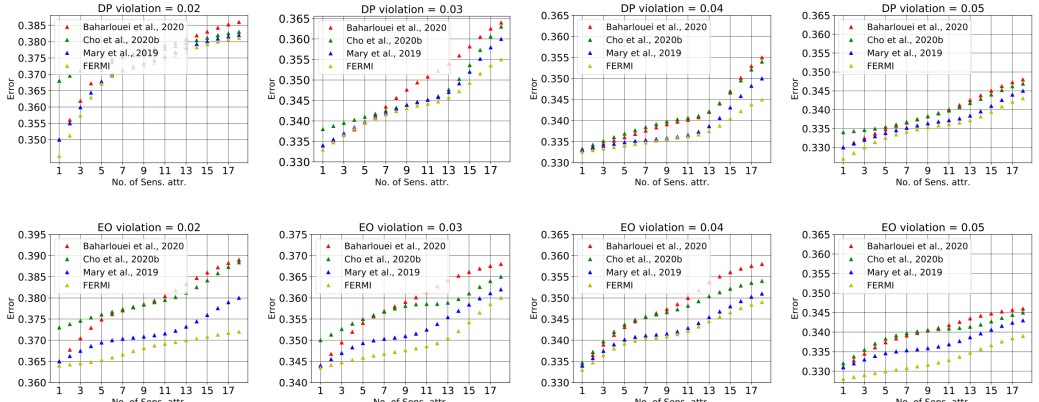

Figure 5: Comparison between FERMI, Mary et al. [2019], Baharlouei et al. [2020], and Cho et al. [2020a] on Communities dataset. Mary et al. [2019] outperforms Baharlouei et al. [2020], Cho et al. [2020a], which we believe could be attributed to the effectiveness of ERMI as a regularizer. FERMI outperforms Mary et al. [2019]. This can be attributed to our empirical formulation Eq. (FERMI obj.) and unbiased stochastic optimization algorithm.

without considering fairness leads to higher false positive rate for the Jew-Muslim group. Figure 4 demonstrates the performance of FERMI, MinDiff Prost et al. [2019], Baharlouei et al. [2020], and naïve baseline on two different batch-sizes: $128$ and $16$. Performance is measured by the overall false positive rate of the trained network and fairness violation is measured by the false positive gap between two sensitive groups (Christians and Jews-Muslims). The network structure is exactly same as the one used by MinDiff Prost et al. [2019]. We can observe that by decreasing the batch size, FERMI maintains the best fairness-accuracy tradeoff compared to other baselines.

### 3.3 Fair Non-binary Classification with Multiple Sensitive Attributes

In this section, we consider a non-binary classification problem with multiple binary sensitive attributes. In this case, we consider the Communities and Crime dataset, which has 18 binary sensitive attributes in total. For our experiments, we pick a subset of $1, 2, 3, \ldots, 18$ sensitive attributes, which corresponds to $|\mathcal{S}| \in \{2, 4, 8, \ldots, 2^{18}\}$. We discretize the target into three classes $\{\text{high}, \text{medium}, \text{low}\}$. The only baselines that we are aware of that can handle non-binary classification with multiple sensitive attributes are Mary et al. [2019], Baharlouei et al. [2020], Cho et al. [2020a], Cho et al. [2020b], and Zhang et al. [2018]. We used the publicly available implementations of Baharlouei et al. [2020] and Cho et al. [2020a] and extended their binary classification algorithms to the non-binary setting.

The results are presented in Fig. 5, where we use conditional demographic parity $L_\infty$ violation (Definition 10) and conditional equal opportunity $L_\infty$ violation (Definition 11) as the fairness violation notions for the two experiments. In each panel, we compare the test error for different number of sensitive attributes for a fixed value of DP violation. It is expected that test error increases with the number of sensitive attributes, as we will have a more stringent fairness constraint to satisfy. As can be seen, compared to the baselines, FERMI offers the most favorable test error vs. fairness violation tradeoffs, particularly as the number of sensitive attributes increases and for the more stringent fairness violation levels, e.g., $0.02$.[10]

### 3.4 Beyond Fairness: Domain Parity Regularization for Domain Generalization

In this section, we demonstrate that our approach may extend beyond fair empirical risk minimization to other problems such as domain generalization. In fact, Li and Vasconcelos [2019], Lahoti et al. [2020], Creager et al. [2021] have already established connections between fair ERM and domain generalization. We consider the Color MNIST dataset Li and Vasconcelos [2019], where all 60,000 training digits are colored with different colors drawn from a class conditional Gaussian distribution

---

[10]Sec. 3.4 demonstrated that using smaller batch sizes results in much more pronounced advantages of FERMI over these baselines.

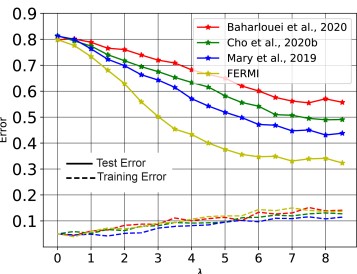 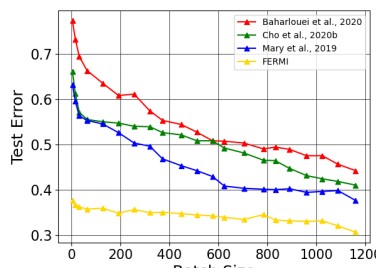

Figure 6: Domain generalization on Color MNIST Li and Vasconcelos [2019] using in-process fair algorithms for demographic parity. **Left panel:** The dashed line is the training error and the solid line is test error. As $\lambda$ increases, fairness regularization results in a learned representation that is less dependent on color; hence training error increases while test error decreases (all algorithms reach a plateau around $\lambda = 8$). We use $|B| = 512$ for all baselines. **Right panel:** We plot test error vs. batch size using an optimized value of $\lambda$ for each algorithm selected via a validation set. The performance of baselines drops 10-20% as batch size becomes small, whereas FERMI is less sensitive to batch size.

with variance $\sigma^2$ around a certain average color for each digit, while the test set remains black and white. Li and Vasconcelos [2019] show that as $\sigma^2 \to 0$, a convolutional network model overfits significantly to each digit's color on the training set, and achieves vanishing training error. However, the learned representation does not generalize to the black and white test set, due to the spurious correlation between digits and color.

Conceptually, the goal of the classifier in this problem is to achieve *high classification accuracy with predictions that are independent of the color of the digit*. We view color as the sensitive attribute in this experiment, and apply fairness baselines for the demographic parity notion of fairness. One would expect that by promoting such independence through a fairness regularizer, generalization would improve (i.e. lower test error on the black and white test set), at the cost of increased training error (on the colored training set). We compare against Mary et al. [2019], Baharlouei et al. [2020], and Cho et al. [2020a] as baselines in this experiment.

The results of this experiment are illustrated in Fig. 6. In the left panel, we see that with no regularization ($\lambda = 0$), the test error is around 80%. As $\lambda$ increases, all methods achieve smaller test error while training error increases. We also observe that *FERMI offers the best test error* in this setup. In the right panel, we observe that decreasing the batch size results in significantly worse generalization for the three baselines considered (due to their biased estimators for the regularizer). However, the negative impact of small batch size is much less severe for FERMI, since FERMI uses unbiased stochastic gradients. In particular, *the performance gap between FERMI and other baselines is more than 20% for $|B| = 64$*. Moreover, *FERMI with minibatch size $|B| = 64$ still outperforms all other baselines with $|B| > 1,000$*. Finally, notice that the test error achieved by FERMI when $\sigma = 0$ is $\sim 30\%$, as compared to more than $50\%$ obtained using REPAIR Li and Vasconcelos [2019] for $\sigma \le 0.05$.

## 4   Discussion and Concluding Remarks

In this paper, we tackled the challenge of developing a fairness-promoting algorithm that is amenable to stochastic optimization. As discussed, algorithms for large-scale ML problems are constrained to use stochastic optimization with (small) minibatches of data in each iteration. To this end, we formulated an empirical objective (FERMI obj.) using ERMI as a regularizer, and derived unbiased stochastic gradient estimators. We proposed the stochastic FERMI algorithm (Algorithm 1) for solving this objective. We then provided the *first theoretical convergence guarantees for a stochastic in-processing fairness algorithm*, by showing that FERMI converges to stationary points of the empirical and population-level objectives (Theorem 1, Theorem 2). Further, these convergence results hold even for non-binary sensitive attributes and non-binary target variables, with any minibatch size.

From an experimental perspective, we showed that *FERMI leads to better fairness-accuracy tradeoffs than all of the state-of-the-art baselines* on a wide variety of binary and non-binary classification tasks (for demographic parity, equalized odds, and equal opportunity). We also showed that these benefits are particularly significant when the number of sensitive attributes grows or the batch size is small. In

particular, we observed that FERMI consistently outperforms Mary et al. [2019] (which tries to solve the same objective Eq. (FRMI obj.)) by up to 20% when the batch size is small. This is not surprising since FERMI is the only algorithm that is guaranteed to find an approximate solution of the fair learning objective with any batch size $|B| \geq 1$. Also, we show in Fig. 7 that the lack of convergence guarantee of Mary et al. [2019] is not just due to more limited analysis: in fact, their stochastic algorithm does not converge. Even in full batch mode, FERMI outperforms all baselines, including Mary et al. [2019] (Fig. 1, Fig. 5). In full batch mode, all baselines should be expected to converge to an approximate solution of their respective empirical objectives, so this suggests that our empirical objective Eq. (FERMI obj.) is fundamentally better, in some sense, than the empirical objectives proposed in prior works. In what sense is Eq. (FERMI obj.) a better empirical objective (apart from permitting stochastic optimization)? For one, it is an asymptotically unbiased estimator of Eq. (FRMI obj.) (by Proposition 2), and Theorem 2 suggests that FERMI algorithm outputs an approximate solution of Eq. (FRMI obj.) for large enough $N$. By contrast, the empirical objectives considered in prior works do not provably yield an approximate solution to the corresponding population-level objective.

The superior fairness-accuracy tradeoffs of FERMI algorithm over the (full batch) baselines also suggests that the underlying population-level objective Eq. (FRMI obj.) has benefits over other fairness objectives. What might these benefits be? First, ERMI upper bounds all other fairness violations (e.g. Shannon mutual information, $L_q$, $L_\infty$) used in the literature: see Appendix C. This implies that ERMI-regularized training yields a model that has small fairness violation with respect to these other notions. Could this also somehow help explain the superior fairness-accuracy tradeoffs achieved by FERMI? Second, the objective function Eq. (FRMI obj.) is easier to optimize than the objectives of competing in-processing methods: ERMI is smooth and is equal to the trace of a matrix (see Lemma 5 in the Appendix), which is easy to compute. Contrast this with the larger computational overhead of Rényi correlation used by Baharlouei et al. [2020], for example, which requires finding the second singular value of a matrix. Perhaps these computational benefits contribute to the observed performance gains? We leave it as future work to rigorously understand the factors that are most responsible for the favorable fairness-accuracy tradeoffs observed from FERMI.

## Broader Impact and Limitations

This paper studied the important problem of developing practical machine learning (ML) algorithms that are *fair* (i.e. non-discriminatory) towards different demographic groups (e.g. race, gender, age). We hope that the societal impacts of our work will be positive, as the deployment of our FERMI algorithm may enable/help companies, government agencies, and other organizations to train large-scale ML models that do not discriminate against certain groups of users. On the other hand, any technology has its limitations, and our algorithm is no exception.

One important limitation of our work is that we have (implicitly) assumed that the data set at hand is labeled accurately and fairly. For example, if race is the sensitive attribute and "likelihood of committing a crime" is the target, then we assume that the training data accurately reflects the criminal histories of all individuals (and in particular does not disproportionately inflate the criminal histories of racial minorities). If this assumption is not satisfied in practice, then the outcomes promoted by our algorithm may not be as fair (in the philosophical sense) as the computed level of fairness violation might suggest. It is even possible that our mitigation strategy could result in more unfairness than unmitigated ERM in this case. More generally, conditional fairness notions like equalized odds suffer from a potential amplification of the inherent discrimination that may exist in the training data. Tackling such issues is beyond the scope of this work; c.f. Kilbertus et al. [2020] and Bechavod et al. [2019].

Another consideration that was not addressed in this paper is the interplay between fairness and other socially consequential AI metrics, such as privacy and robustness (e.g. to data poisoning). It is possible that our algorithm could *leak* sensitive information about individuals in the training data set (e.g. via membership inference attacks or model inversion attacks), even if the data is anonymous Fredrikson et al. [2015], Shokri et al. [2017], Faizullabhoy and Korolova [2018], Nasr et al. [2019], Carlini et al. [2021]. Such leaks could be used maliciously, e.g. to discriminate against an individual with a confidential disability. *Differential privacy* Dwork et al. [2006] ensures that sensitive data cannot be leaked (with high probability), and the interplay between fairness and privacy has been explored (see e.g. Jagielski et al. [2019], Xu et al. [2019], Cummings et al. [2019], Mozannar et al. [2020], Tran et al. [2021a,b]. Developing and analyzing a differentially private version of

FERMI could be an interesting direction for future work. Another potential threat to FERMI-trained models is data poisoning attacks. While our experiments demonstrated that FERMI is relatively effective with missing sensitive attributes, we did not investigate its performance in the presence of label flipping or other poisoning attacks. Exploring and improving the robustness of FERMI is another avenue for future research.

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

# Appendix

We provide a table of contents below for easier navigation of the appendix.

**CONTENTS**

# A Notions of Fairness

Let $(Y, \widehat{Y}, \mathcal{A}, S)$ denote the true target, predicted target, the advantaged outcome class, and the sensitive attribute, respectively. We review three major notions of fairness.

**Definition 2** (demographic parity Dwork et al. [2012])**.** *We say that a learning machine satisfies demographic parity if $\widehat{Y}$ is independent of $S$.*

**Definition 3** (equalized odds Hardt et al. [2016])**.** *We say that a learning machine satisfies equalized odds, if $\widehat{Y}$ is conditionally independent of $S$ given $Y$.*

**Definition 4** (equal opportunity Hardt et al. [2016])**.** *We say that a learning machine satisfies equal opportunity with respect to $\mathcal{A}$, if $\widehat{Y}$ is conditionally independent of $S$ given $Y = y$ for all $y \in \mathcal{A}$.*

Notice that the equal opportunity as defined here generalizes the definition in Hardt et al. [2016]. It recovers equalized odds if $\mathcal{A} = \mathcal{Y}$, and it recovers equal opportunity of Hardt et al. [2016] for $\mathcal{A} = \{1\}$ in binary classification.

# B   ERMI: General Definition, Properties, and Special Cases Unraveled

We begin by stating a notion of fairness that generalizes demographic parity, equalized odds, and equal opportunity fairness definitions (the three notions considered in this paper). This will be convenient for defining ERMI in its general form and presenting the results in Appendix C. Consider a learner who trains a model to make a prediction, $\widehat{Y}$, e.g., whether or not to extend a loan, supported on a set $\mathcal{Y}$. Here we allow $\widehat{Y}$ to be either discrete or continuous. The prediction is made using a set of features, $\mathbf{X}$, e.g., financial history features. We assume that there is a set of discrete sensitive attributes, $S$, e.g., race and sex, supported on $\mathcal{S}$, associated with each sample. Further, let $\mathcal{A} \subseteq \mathcal{Y}$ denote an advantaged outcome class, e.g., the outcome where a loan is extended.

**Definition 5** (($Z, \mathcal{Z}$)-fairness)**.** *Given a random variable $Z$, let $\mathcal{Z}$ be a subset of values that $Z$ can take. We say that a learning machine satisfies ($Z, \mathcal{Z}$)-fairness if for every $z \in \mathcal{Z}$, $\widehat{Y}$ is conditionally independent of $S$ given $Z = z$, i.e. $\forall \hat{y} \in \mathcal{Y}, s \in \mathcal{S}, z \in \mathcal{Z}, p_{\widehat{Y},S|Z}(\hat{y}, s|z) = p_{\widehat{Y}|Z}(\hat{y}|z)p_{S|Z}(s|z)$.*

($Z, \mathcal{Z}$)-fairness includes the popular demographic parity, equalized odds, and equal opportunity notions of fairness as special cases:

1. ($Z, \mathcal{Z}$)-fairness recovers demographic parity Dwork et al. [2012] if $Z = 0$ and $\mathcal{Z} = \{0\}$. In this case, conditioning on $Z$ has no effect, and hence $(0, \{0\})$ fairness is equivalent to the independence between $\widehat{Y}$ and $S$ (see Definition 2, Appendix A).

2. ($Z, \mathcal{Z}$)-fairness recovers equalized odds Hardt et al. [2016] if $Z = Y$ and $\mathcal{Z} = \mathcal{Y}$. In this case, $Z \in \mathcal{Z}$ is trivially satisfied. Hence, conditioning on $Z$ is equivalent to conditioning on $Y$, which recovers the equalized odds notion of fairness, i.e., conditional independence of $\widehat{Y}$ and $S$ given $Y$ (see Definition 3, Appendix A).

3. ($Z, \mathcal{Z}$)-fairness recovers equal opportunity Hardt et al. [2016] if $Z = Y$ and $\mathcal{Z} = \mathcal{A}$. This is also similar to the previous case with $\mathcal{Y}$ replaced with $\mathcal{A}$ (see Definition 4, Appendix A).

Note that verifying ($Z, \mathcal{Z}$)-fairness requires having access to the joint distribution of random variables $(Z, \widehat{Y}, S)$. This joint distribution is unavailable to the learner in the context of machine learning, and hence the learner would resort to empirical estimation of the amount of violation of independence, measured through some divergence. See Williamson and Menon [2019] for a related discussion.

In this general context, here is the general definition of ERMI:

**Definition 6** (ERMI – exponential Rényi mutual information)**.** *We define the exponential Rényi mutual information between $\widehat{Y}$ and $S$ given $Z \in \mathcal{Z}$ as*

$$D_R(\widehat{Y}; S|Z \in \mathcal{Z}) := \mathbb{E}_{Z,\widehat{Y},S} \left\{ \frac{p_{\widehat{Y},S|Z}(\widehat{Y}, S|Z)}{p_{\widehat{Y}|Z}(\widehat{Y}|Z)p_{S|Z}(S|Z)} \middle| Z \in \mathcal{Z} \right\} - 1. \qquad \text{(ERMI)}$$

Notice that ERMI is in fact the $\chi^2$-divergence between the conditional joint distribution, $p_{\widehat{Y},S}$, and the Kronecker product of conditional marginals, $p_{\widehat{Y}} \otimes p_S$, where the conditioning is on $Z \in \mathcal{Z}$. Further, $\chi^2$-divergence is an $f$-divergence with $f(t) = (t-1)^2$. See [Csiszár and Shields, 2004, Section 4] for a discussion. As an immediate result of this observation and well-known properties of $f$-divergences, we can state the following property of ERMI:

**Remark 2.** $D_R(\widehat{Y}; S|Z \in \mathcal{Z}) \geq 0$ *with equality if and only if for all $z \in \mathcal{Z}$, $\widehat{Y}$ and $S$ are conditionally independent given $Z = z$.*

To further clarify the definition of ERMI, especially as it relates to demographic parity, equalized odds, and equal opportunity, we will unravel the definition explicitly in a few special cases.

First, let $Z = 0$ and $\mathcal{Z} = \{0\}$. In this case, $Z \in \mathcal{Z}$ trivially holds, and conditioning on $Z$ has no effect, resulting in:

$$
\begin{aligned}
D_R(\widehat{Y}; S) &:= D_R(\widehat{Y}; S | Z \in \mathcal{Z})\Big|_{Z=0, \mathcal{Z}=\{0\}} \\
&= \mathbb{E}_{\widehat{Y}, S}\left\{ \frac{p_{\widehat{Y}, S}(\widehat{Y}, S)}{p_{\widehat{Y}}(\widehat{Y}) p_S(S)} \right\} - 1 \\
&= \sum_{s \in \mathcal{S}} \int_{\hat{y} \in \mathcal{Y}} \frac{p_{\widehat{Y}, S}(\hat{y}, s) - p_{\widehat{Y}}(\hat{y}) p_S(s)}{p_{\widehat{Y}}(\hat{y}) p_S(s)} p_{\widehat{Y}, S}(\hat{y}, s) d\hat{y}.
\end{aligned}
\tag{2}
$$

$D_R(\widehat{Y}; S)$ is the notion of ERMI that should be used when the desired notion of fairness is demographic parity. In particular, $D_R(\widehat{Y}; S) = 0$ implies that $\chi^2$ divergence between $p_{\widehat{Y}, S}$, and the Kronecker product of marginals, $p_{\widehat{Y}} \otimes p_S$ is zero. This in turn implies that $\widehat{Y}$ and $S$ are independent, which is the definition of demographic parity. We note that when $\widehat{Y}$ and $S$ are discrete, this special case ($Z = 0$ and $\mathcal{Z} = \{0\}$) of ERMI is referred to as $\chi^2$-information by Calmon et al. [2017b].

Next, we consider $Z = Y$ and $\mathcal{Z} = \mathcal{Y}$. In this case, $Z \in \mathcal{Z}$ is trivially satisfied, and hence,

$$
\begin{aligned}
D_R(\widehat{Y}; S | Y) &:= D_R(\widehat{Y}; S | Z \in \mathcal{Z})\Big|_{Z=Y, \mathcal{Z}=\mathcal{Y}} \\
&= \mathbb{E}_{Y, \widehat{Y}, S}\left\{ \frac{p_{\widehat{Y}, S | Y}(\widehat{Y}, S | Y)}{p_{\widehat{Y} | Y}(\widehat{Y} | Y) p_{S | Y}(S | Y)} \right\} - 1 \\
&= \sum_{s \in \mathcal{S}} \int_{y \in \mathcal{Y}} \int_{\hat{y} \in \mathcal{Y}} \frac{p_{\widehat{Y}, S | Y}(\hat{y}, s | y) - p_{\widehat{Y} | Y}(\hat{y} | y) p_{S | Y}(s | y)}{p_{\widehat{Y} | Y}(\hat{y} | y) p_{S | Y}(s | y)} p_{Y, \widehat{Y}, S}(y, \hat{y}, s) d\hat{y} dy \\
&= \sum_{s \in \mathcal{S}} \int_{y \in \mathcal{Y}} \int_{\hat{y} \in \mathcal{Y}} \frac{p_{\widehat{Y}, S | Y}(\hat{y}, s | y)^2}{p_{\widehat{Y} | Y}(\hat{y} | y) p_{S | Y}(s | y)} p_Y(y) d\hat{y} dy - 1.
\end{aligned}
\tag{3}
$$

$D_R(\widehat{Y}; S | Y)$ should be used when the desired notion of fairness is equalized odds. In particular, $D_R(\widehat{Y}; S | Y) = 0$ directly implies the conditional independence of $\widehat{Y}$ and $S$ given $Y$.

Finally, we consider $Z = Y$ and $\mathcal{Z} = \mathcal{A}$. In this case, we have

$$
\begin{aligned}
D_R^{\mathcal{A}}(\widehat{Y}; S | Y) &:= D_R(\widehat{Y}; S | Z \in \mathcal{Z})\Big|_{Z=Y, \mathcal{Z}=\mathcal{A}} \\
&= \mathbb{E}_{Y, \widehat{Y}, S}\left\{ \frac{p_{\widehat{Y}, S | Y}(\widehat{Y}, S | Y)}{p_{\widehat{Y} | Y}(\widehat{Y} | Y) p_{S | Y}(S | Y)} \Bigg| Y \in \mathcal{A} \right\} - 1 \\
&= \sum_{s \in \mathcal{S}} \int_{y \in \mathcal{A}} \int_{\hat{y} \in \mathcal{Y}} \frac{p_{\widehat{Y}, S | Y}(\hat{y}, s | y) - p_{\widehat{Y} | Y}(\hat{y} | y) p_{S | Y}(s | y)}{p_{\widehat{Y} | Y}(\hat{y} | y) p_{S | Y}(s | y)} p_Y^{\mathcal{A}}(y) d\hat{y} dy \\
&= \sum_{s \in \mathcal{S}} \int_{y \in \mathcal{A}} \int_{\hat{y} \in \mathcal{Y}} \frac{p_{\widehat{Y}, S | Y}(\hat{y}, s | y)^2}{p_{\widehat{Y} | Y}(\hat{y} | y) p_{S | Y}(s | y)} p_{\widehat{Y}, S | Y}(\hat{y}, s | y) p_Y^{\mathcal{A}}(y) d\hat{y} dy - 1,
\end{aligned}
\tag{4}
$$

where

$$
p_Y^{\mathcal{A}}(y) := \frac{p_Y(y)}{\int_{y' \in \mathcal{A}} p_Y(y') dy'}.
\tag{5}
$$

This notion is what should be used when the desired notion of fairness is equal opportunity. This can be further simplified when the advantaged class is a singleton (which is the case in binary

classification). If $Z = Y$ and $\mathcal{Z} = \{y\}$, then

$$
\begin{aligned}
D_R(\widehat{Y}; S|Y = y) &:= D_R^{\{y\}}(\widehat{Y}; S|Y) \\
&= \sum_{s \in \mathcal{S}} \int_{\hat{y} \in \mathcal{Y}} \frac{p_{\widehat{Y},S|Y}(\hat{y}, s|y) - p_{\widehat{Y}|Y}(\hat{y}|y)p_{S|Y}(s|y)}{p_{\widehat{Y}|Y}(\hat{y}|y)p_{S|Y}(s|y)} p_{\widehat{Y},S|Y}(\hat{y}, s|y) d\hat{y} \\
&= \sum_{s \in \mathcal{S}} \int_{\hat{y} \in \mathcal{Y}} \frac{p_{\widehat{Y},S|Y}(\hat{y}, s|y)^2}{p_{\widehat{Y}|Y}(\hat{y}|y)p_{S|Y}(s|y)} d\hat{y} - 1.
\end{aligned}
\tag{6}
$$

Finally, we note that we use the notation $D_R(\widehat{Y}; S|Y)$ and $D_R(\widehat{Y}; S|Y = y)$ to be consistent with the definition of conditional mutual information in Cover and Thomas [1991].

# C   Relations Between ERMI and Other Fairness Violation Notions

Recall that most existing in-processing methods use some notion of fairness violation as a regularizer to enforce fairness in the trained model. These notions of fairness violation typically take the form of some information divergence between the sensitive attributes and the predicted targets (e.g. Mary et al. [2019], Baharlouei et al. [2020], Cho et al. [2020b]). In this section, we show that ERMI provides an upper bound on all of the existing measures of fairness violations for demographic parity, equal opportunity, and equalized odds. As mentioned in the main body, this insight might help explain the favorable empirical performance of our algorithm compared to baselines–even when full batch is used. In particular, the results in this section imply that FERMI algorithm leads to small fairness violation with respect to ERMI and all of these other measures.

We should mention that many of these properties of $f$ divergences are well-known or easily derived from existing results, so we do not intend to claim great originality with any of these results. That said, we include proofs of all results for which we are not aware of any references with proofs. The results in this section also hold for continuous (or discrete) $\widehat{Y}$. We will now state and discuss these results before proving them.

**Definition 7** (Rényi mutual information Rényi [1961])**.** *Let the Rényi mutual information of order $\alpha > 1$ between random variables $\widehat{Y}$ and $S$ given $Z \in \mathcal{Z}$ be defined as:*

$$I_\alpha(\widehat{Y}; S | Z \in \mathcal{Z}) := \frac{1}{\alpha - 1} \log \left( \mathbb{E}_{Z, \widehat{Y}, S} \left\{ \left( \frac{p_{\widehat{Y}, S | Z}(\widehat{Y}, S | Z)}{p_{\widehat{Y} | Z}(\widehat{Y} | Z) p_{S | Z}(S | Z)} \right)^{\alpha - 1} \middle| Z \in \mathcal{Z} \right\} \right), \qquad \text{(RMI)}$$

*which generalizes Shannon mutual information*

$$I_1(\widehat{Y}; S | Z \in \mathcal{Z}) := \mathbb{E}_{Z, \widehat{Y}, S} \left\{ \log \left( \frac{p_{\widehat{Y}, S | Z}(\widehat{Y}, S | Z)}{p_{\widehat{Y} | Z}(\widehat{Y} | Z) p_{S | Z}(S | Z)} \right) \middle| Z \in \mathcal{Z} \right\}, \qquad \text{(MI)}$$

*and recovers it as $\lim_{\alpha \to 1^+} I_\alpha(\widehat{Y}; S | Z \in \mathcal{Z}) = I_1(\widehat{Y}; S | Z \in \mathcal{Z})$.*

Note that $I_\alpha(\widehat{Y}; S | Z \in \mathcal{Z}) \geq 0$ with equality if and only if $(Z, \mathcal{Z})$-fairness is satisfied.

The following is a minor change from results in Sason and Verdú [2016]:

**Lemma 1** (ERMI provides an upper bound for Shannon mutual information)**.** *We have*

$$0 \leq I_1(\widehat{Y}; S | Z \in \mathcal{Z}) \leq I_2(\widehat{Y}; S | Z \in \mathcal{Z}) \leq e^{I_2(\widehat{Y}; S | Z \in \mathcal{Z})} - 1 = D_R(\widehat{Y}; S | Z \in \mathcal{Z}). \qquad (7)$$

Lemma 1 also shows that ERMI is exponentially related to the Rényi mutual information of order 2. We include a proof below for completeness.

**Definition 8** (Rényi correlation Hirschfeld [1935], Gebelein [1941], Rényi [1959])**.** *Let $\mathcal{F}$ and $\mathcal{G}$ be the set of measurable functions such that for random variables $\widehat{Y}$ and $S$, $\mathbb{E}_{\widehat{Y}}\{f(\widehat{Y}; z)\} = \mathbb{E}_S\{g(S; z)\} = 0$, $\mathbb{E}_{\widehat{Y}}\{f(\widehat{Y}; z)^2\} = \mathbb{E}_S\{g(S; z)^2\} = 1$, for all $z \in \mathcal{Z}$. Rényi correlation is:*

$$\rho_R(\widehat{Y}, S | Z \in \mathcal{Z}) := \sup_{f, g \in \mathcal{F} \times \mathcal{G}} \mathbb{E}_{Z, \widehat{Y}, S} \left\{ f(\widehat{Y}; Z) g(S; Z) \middle| Z \in \mathcal{Z} \right\}. \qquad \text{(RC)}$$

Rényi correlation generalizes Pearson correlation,

$$\rho(\widehat{Y}, S | Z \in \mathcal{Z}) := \mathbb{E}_Z \left\{ \frac{\mathbb{E}_{\widehat{Y}, S}\{\widehat{Y} S | Z\}}{\sqrt{\mathbb{E}_{\widehat{Y}}\{\widehat{Y}^2 | Z\} \mathbb{E}_S\{S^2 | Z\}}} \middle| Z \in \mathcal{Z} \right\}, \qquad \text{(PC)}$$

to capture nonlinear dependencies between the random variables by finding functions of random variables that maximize the Pearson correlation coefficient between the random variables. In fact, it is true that $\rho_R(\widehat{Y}, S | Z \in \mathcal{Z}) \geq 0$ with equality if and only if $(Z, \mathcal{Z})$-fairness is satisfied. Rényi correlation has gained popularity as a measure of fairness violation Mary et al. [2019], Baharlouei et al. [2020], Grari et al. [2020]. Rényi correlation is also upper bounded by ERMI. The following result has already been shown by Mary et al. [2019] and we present it for completeness.

**Lemma 2** (ERMI provides an upper bound for Rényi correlation). *We have*

$$0 \leq |\rho(\widehat{Y}, S|Z \in \mathcal{Z})| \leq \rho_R(\widehat{Y}, S|Z \in \mathcal{Z}) \leq D_R(\widehat{Y}; S|Z \in \mathcal{Z}), \tag{8}$$

*and if $|\mathcal{S}| = 2$, $D_R(\widehat{Y}; S|Z \in \mathcal{Z}) = \rho_R(\widehat{Y}, S|Z \in \mathcal{Z})$.*

**Definition 9** ($L_q$ fairness violation). *We define the $L_q$ fairness violation for $q \geq 1$ by:*

$$L_q(\widehat{Y}, S|Z \in \mathcal{Z}) := \mathbb{E}_Z \left\{ \left( \int_{\hat{y} \in \mathcal{Y}_0} \sum_{s \in \mathcal{S}_0} \left| p_{\widehat{Y},S|Z}(\hat{y}, s|Z) - p_{\widehat{Y}|Z}(\hat{y}|Z) p_{S|Z}(s|Z) \right|^q dy \right)^{\frac{1}{q}} \middle| Z \in \mathcal{Z} \right\}. \tag{Lq}$$

Note that $L_q(\widehat{Y}, S|Z \in \mathcal{Z}) = 0$ if and only if $(Z, \mathcal{Z})$-fairness is satisfied. In particular, $L_\infty$ fairness violation recovers demographic parity violation [Kearns et al., 2018, Definition 2.1] if we let $\mathcal{Z} = \{0\}$ and $Z = 0$. It also recovers equal opportunity violation Hardt et al. [2016] if $\mathcal{Z} = \mathcal{A}$ and $Z = Y$.

**Lemma 3** (ERMI provides an upper bound for $L_\infty$ fairness violation). *Let $\widehat{Y}$ be a discrete or continuous random variable, and $S$ be a discrete random variable supported on a finite set. Then for any $q \geq 1$,*

$$0 \leq L_q(\widehat{Y}, S|Z \in \mathcal{Z}) \leq \sqrt{D_R(\widehat{Y}, S|Z \in \mathcal{Z})}. \tag{9}$$

The above lemma says that if a method controls ERMI value for imposing fairness, then $L_\infty$ violation is controlled. In particular, the variant of ERMI that is specialized to demographic parity also controls $L_\infty$ demographic parity violation Kearns et al. [2018]. The variant of ERMI that is specialized to equal opportunity also controls the $L_\infty$ equal opportunity violation Hardt et al. [2016]. While our algorithm uses ERMI as a regularizer, in our experiments, we measure fairness violation through the more commonly used $L_\infty$ violation. Despite this, we show that our approach leads to better tradeoff curves between fairness violation and performance.

**Remark.** The bounds in Lemmas 1-3 are not tight in general, but this is not of practical concern. They show that bounding ERMI is sufficient because any model that achieves small ERMI is guaranteed to satisfy any other fairness violation. This makes ERMI an effective regularizer for promoting fairness. In fact, in Sec. 3, we saw that our algorithm, FERMI, achieves the best tradeoffs between fairness violation and performance across state-of-the-art baselines.

*Proof of Lemma 1.* We proceed to prove all the (in)equalities one by one:

- $0 \leq I_S(\widehat{Y}; S|Z \in \mathcal{Z})$. This is well known and the proof can be found in any information theory textbook Cover and Thomas [1991].

- $I_1(\widehat{Y}; S|Z \in \mathcal{Z}) \leq I_2(\widehat{Y}; S|Z \in \mathcal{Z})$. This is a known property of Rényi mutual information, but we provide a proof for completeness in Lemma 4 below.

- $I_2(\widehat{Y}; S|Z \in \mathcal{Z}) \leq e^{I_2(\widehat{Y}; S|Z \in \mathcal{Z})} - 1$. This follows from the fact that $x \leq e^x - 1$.

- $e^{I_2(\widehat{Y}; S)|Z \in \mathcal{Z}} - 1 = D_R(\widehat{Y}; S|Z \in \mathcal{Z})$. This follows from simple algebraic manipulation.

$\square$

**Lemma 4.** *Let $\widehat{Y}, S, Z$ be discrete or continuous random variables. Then:*

(a) *For any $\alpha, \beta \in [1, \infty]$, $I_\beta(\widehat{Y}; S|Z \in \mathcal{Z}) \geq I_\alpha(\widehat{Y}; S|Z \in \mathcal{Z})$ if $\beta > \alpha$.*

(b) $\lim_{\alpha \to 1^+} I_\alpha(\widehat{Y}; S|Z \in \mathcal{Z}) = I_1(\widehat{Y}; S) := \mathbb{E}_Z \left\{ D_{KL}(p_{\widehat{Y},S|Z} || p_{\widehat{Y}|Z} \otimes p_{S|Z}) \middle| Z \in \mathcal{Z} \right\}$, *where $I_1(\cdot; \cdot)$ denotes the Shannon mutual information and $D_{KL}$ is Kullback-Leibler divergence (relative entropy).*

(c) *For all $\alpha \in [1, \infty]$, $I_\alpha(\widehat{Y}; S|Z \in \mathcal{Z}) \geq 0$ with equality if and only if for all $z \in \mathcal{Z}$, $\widehat{Y}$ and $S$ are conditionally independent given $z$.*

*Proof. (a)* First assume $0 < \alpha < \beta < \infty$ and that $\alpha, \beta \neq 1$. Define $a = \alpha - 1$, and $b = \beta - 1$. Then the function $\phi(t) = t^{b/a}$ is convex for all $t \geq 0$, so by Jensen's inequality we have:

$$\frac{1}{b} \log \left( \mathbb{E} \left\{ \left( \frac{p(\widehat{Y}, S|Z)}{p(\widehat{Y}|Z)p(S|Z)} \right)^b \middle| Z \in \mathcal{Z} \right\} \right) \geq \frac{1}{b} \log \left( \mathbb{E} \left\{ \left( \frac{p(\widehat{Y}, S|Z)}{p(\widehat{Y}|Z)p(S|Z)} \right)^a \middle| Z \in \mathcal{Z} \right\}^{b/a} \right)$$

$$= \frac{1}{a} \log \left( \mathbb{E} \left\{ \left( \frac{p(\widehat{Y}, S|Z)}{p(\widehat{Y}|Z)p(S|Z)} \right)^a \middle| Z \in \mathcal{Z} \right\} \right). \tag{10}$$

Now suppose $\alpha = 1$. Then by the monotonicity for $\alpha \neq 1$ proved above, we have $I_1(\widehat{Y}; S) = \lim_{\alpha \to 1^-} I_\alpha(\widehat{Y}; S) = \sup_{\alpha \in (0,1)} I_\alpha(\widehat{Y}; S) \leq \inf_{\alpha > 1} I_\alpha(\widehat{Y}; S)$. Also, $I_\infty(\widehat{Y}; S) = \lim_{\alpha \to \infty} I_\alpha(\widehat{Y}; S) = \sup_{\alpha > 0} I_\alpha(\widehat{Y}; S)$.

*(b)* This is a standard property of the cumulant generating function (see Dembo and Zeitouni [2009]).

*(c)* It is straightforward to observe that independence implies that Rényi mutual information vanishes. On the other hand, if Rényi mutual information vanishes, then part (a) implies that Shannon mutual information also vanishes, which implies the desired conditional independence. □

*Proof of Lemma 2.* The proof is completed using the following pieces.

- $0 \leq |\rho(\widehat{Y}, S|Z \in \mathcal{Z})| \leq \rho_R(\widehat{Y}, S|Z \in \mathcal{Z})$. This is obvious from the definition of $\rho_R(\widehat{Y}, S|Z \in \mathcal{Z})$.

- $\rho_R(\widehat{Y}, S|Z \in \mathcal{Z}) \leq D_R(\widehat{Y}; S|Z \in \mathcal{Z})$. This follows from Lemma 5 below.

- Notice that if $|\mathcal{S}| = 2$, Lemma 5 implies that $D_R(\widehat{Y}; S|Z \in \mathcal{Z}) = \rho_R(\widehat{Y}, S|Z \in \mathcal{Z})$.

□

Next, we recall the following lemma, which is stated in Mary et al. [2019] and derives from Witsenhausen's characterization of Renyi correlation:

**Lemma 5.** *Suppose that $\mathcal{S} = [k]$. Let the $k \times k$ matrix $P$ be defined as $P = \{P_{ij}\}_{i,j \in [k] \times [k]}$, where*

$$P_{ij} := \frac{1}{\sqrt{p_S(i)p_S(j)}} \int_{y \in \mathcal{Y}} \left( \frac{p_{\widehat{Y},S}(y,i) p_{\widehat{Y},S}(y,j)}{p_{\widehat{Y}}(y)} \right) dy. \tag{11}$$

*Let $1 = \sigma_1 \geq \sigma_2 \geq \ldots \geq \sigma_k \geq 0$ be the eigenvalues of $P$. Then,*

$$\rho_R(\widehat{Y}, S) = \sigma_2, \tag{12}$$

$$D_R(\widehat{Y}; S) = \mathrm{Tr}(P) - 1 = \sum_{i=2}^{k} \sigma_i. \tag{13}$$

*Proof.* Eq. (12) is proved in [Witsenhausen, 1975, Section 3]. To prove Eq. (13), notice that

$$\mathrm{Tr}(P) = \sum_{i \in [k]} P_{ii}$$

$$= \sum_{i \in [k]} \frac{1}{p_S(i)} \int_{y \in \mathcal{Y}} \left( \frac{p_{\widehat{Y},S}(y,i)^2}{p_{\widehat{Y}}(y)} \right) dy$$

$$= E_{\widehat{Y},S} \left\{ \left( \frac{p_{\widehat{Y},S}(\widehat{Y}, S)}{p_{\widehat{Y}}(\widehat{Y}) p_S(S)} \right) \right\}$$

$$= 1 + D_R(\widehat{Y}; S),$$

which completes the proof. □

*Proof of Lemma 3.* It suffices to prove the inequality for $L_1$, as $L_q$ is bounded above by $L_1$ for all $q \geq 1$. The proof for the case where $Z = 0$ and $\mathcal{Z} = \{0\}$ follows from the following set of inequalities:

$$L_1(\widehat{Y}, S | Z \in \mathcal{Z}) = \sum_{s \in \mathcal{S}} \int_{y \in \mathcal{Y}} \left| p_{\widehat{Y},S}(y,s) - p_{\widehat{Y}}(y) p_S(s) \right| dy \tag{14}$$

$$= \sum_{s \in \mathcal{S}} \int_{y \in \mathcal{Y}} \sqrt{p_{\widehat{Y}}(y) p_S(s)} \frac{\left| p_{\widehat{Y},S}(y,s) - p_{\widehat{Y}}(y) p_S(s) \right|}{\sqrt{p_{\widehat{Y}}(y) p_S(s)}} dy \tag{15}$$

$$\leq \sqrt{\left( \sum_{s \in \mathcal{S}} \int_{y \in \mathcal{Y}} p_{\widehat{Y}}(y) p_S(s) dy \right) \left( \sum_{s \in \mathcal{S}} \int_{y \in \mathcal{Y}} \left( \frac{(p_{\widehat{Y},S}(y,s) - p_{\widehat{Y}}(y) p_S(s))^2}{p_{\widehat{Y}}(y) p_S(s)} \right) \right)} \tag{16}$$

$$\leq \sqrt{\sum_{s \in \mathcal{S}} \int_{y \in \mathcal{Y}} \left( \frac{(p_{\widehat{Y},S}(y,s) - p_{\widehat{Y}}(y) p_S(s))^2}{p_{\widehat{Y}}(y) p_S(s)} \right) dy} \tag{17}$$

$$= \sqrt{D_R(\widehat{Y}; S)}, \tag{18}$$

where Eq. (16) follows from Cauchy-Schwarz inequality, and Eq. (18) follows from Lemma 6. The extension to general $Z$ and $\mathcal{Z}$ is immediate by observing that $\rho(\widehat{Y}, S | Z \in \mathcal{Z}) = \mathbb{E}_Z \left[ \rho(\widehat{Y}, S | Z) \mid Z \in \mathcal{Z} \right]$, $\rho_R(\widehat{Y}, S | Z \in \mathcal{Z}) = \mathbb{E}_Z \left[ \rho_R(\widehat{Y}, S | Z) \mid Z \in \mathcal{Z} \right]$, and $D_R(\widehat{Y}, S | Z \in \mathcal{Z}) = \mathbb{E}_Z \left[ D_R(\widehat{Y}, S | Z) \mid Z \in \mathcal{Z} \right]$.

$\square$

**Lemma 6.** *We have*

$$D_R(\widehat{Y}; S) = \sum_{s \in \mathcal{S}} \int_{y \in \mathcal{Y}} \left( \frac{(p_{\widehat{Y},S}(y,s) - p_{\widehat{Y}}(y) p_S(s))^2}{p_{\widehat{Y}}(y) p_S(s)} \right) dy. \tag{19}$$

*Proof.* The proof follows from the following set of identities:

$$\sum_{s \in \mathcal{S}} \int_{y \in \mathcal{Y}} \left( \frac{(p_{\widehat{Y},S}(y,s) - p_{\widehat{Y}}(y) p_S(s))^2}{p_{\widehat{Y}}(y) p_S(s)} \right) dy = \sum_{s \in \mathcal{S}} \int_{y \in \mathcal{Y}} \frac{(p_{\widehat{Y},S}(y,s))^2}{p_{\widehat{Y}}(y) p_S(s)} dy$$

$$- 2 \sum_{s \in \mathcal{S}} \int_{y \in \mathcal{Y}} p_{\widehat{Y},S}(y,s) dy$$

$$+ \sum_{s \in \mathcal{S}} \int_{y \in \mathcal{Y}} p_{\widehat{Y}}(y) p_S(s) dy \tag{20}$$

$$= E \left\{ \frac{p_{\widehat{Y},S}(\widehat{Y}, S)}{p_{\widehat{Y}}(\widehat{Y}) p_S(S)} \right\} - 1 \tag{21}$$

$$= D_R(\widehat{Y}; S). \tag{22}$$

$\square$

Next, we present some alternative fairness definitions and show that they are also upper bounded by ERMI.

**Definition 10** (conditional demographic parity $L_\infty$ violation). *Given a predictor $\widehat{Y}$ supported on $\mathcal{Y}$ and a discrete sensitive attribute $S$ supported on a finite set $\mathcal{S}$, we define the conditional demographic parity violation by:*

$$\widetilde{dp}(\widehat{Y}|S) := \sup_{\widehat{y} \in \mathcal{Y}} \max_{s \in \mathcal{S}} \left| p_{\widehat{Y}|S}(\widehat{y}|s) - p_{\widehat{Y}}(\widehat{y}) \right|. \tag{23}$$

First, we show that $\widetilde{dp}(\widehat{Y}|S)$ is a reasonable notion of fairness violation.

**Lemma 7.** $\widetilde{dp}(\widehat{Y}|S) = 0$ *iff (if and only if) $\widehat{Y}$ and $S$ are independent.*

*Proof.* By definition, $\widetilde{dp}(\widehat{Y}|S) = 0$ iff for all $\widehat{y} \in \mathcal{Y}, s \in \mathcal{S}, p_{\widehat{Y},S}(\widehat{y}|s) = p_{\widehat{Y}}(\widehat{y})$ iff $\widehat{Y}$ and $S$ are independent (since we always assume $p(s) > 0$ for all $s \in \mathcal{S}$). $\qquad\square$

**Lemma 8** (ERMI provides an upper bound for conditional demographic parity $L_\infty$ violation)**.** *Let $\widehat{Y}$ be a discrete or continuous random variable supported on $\mathcal{Y}$, and $S$ be a discrete random variable supported on a finite set $\mathcal{S}$. Denote $p_S^{\min} := \min_{s\in\mathcal{S}} p_S(s) > 0$. Then,*

$$0 \leq \widetilde{dp}(\widehat{Y}|S) \leq \frac{1}{p_S^{\min}}\sqrt{D_R(\widehat{Y};S)}. \tag{24}$$

*Proof.* The proof follows from the following set of (in)equalities:

$$\left(\widetilde{dp}(\widehat{Y}|S)\right)^2 = \sup_{\widehat{y}\in\mathcal{Y}} \max_{s\in\mathcal{S}} \left(p_{\widehat{Y}|S}(\widehat{y}|s) - p_{\widehat{Y}}(\widehat{y})\right)^2 \tag{25}$$

$$\leq \frac{1}{(p_S^{\min})^2} \sup_{\widehat{y}\in\mathcal{Y}} \max_{s\in\mathcal{S}} \left(p_{\widehat{Y},S}(\widehat{y},s) - p_{\widehat{Y}}(\widehat{y})p_S(s)\right)^2 \tag{26}$$

$$\leq \frac{1}{(p_S^{\min})^2} \int_{\widehat{y}\in\mathcal{Y}} \sum_{s\in\mathcal{S}} \left(p_{\widehat{Y},S}(\widehat{y},s) - p_{\widehat{Y}}(\widehat{y})p_S(s)\right)^2 \tag{27}$$

$$= \frac{1}{(p_S^{\min})^2} D_R(\widehat{Y};S), \tag{28}$$

where Eq. (28) follows from Lemma 3. $\qquad\square$

**Definition 11** (conditional equal opportunity $L_\infty$ violation Hardt et al. [2016])**.** *Let $Y, \widehat{Y}$ take values in $\mathcal{Y}$ and let $\mathcal{A} \subseteq \mathcal{Y}$ be a compact subset denoting the advantaged outcomes (For example, the decision "to interview" an individual or classify an individual as a "low risk" for financial purposes). We define the conditional equal opportunity $L_\infty$ violation of $\widehat{Y}$ with respect to the sensitive attribute $S$ and the advantaged outcome $\mathcal{A}$ by*

$$\widetilde{eo}(\widehat{Y}|S, Y \in \mathcal{A}) := \mathbb{E}_Y \left\{ \sup_{\widehat{y}\in\mathcal{Y}} \max_{s\in\mathcal{S}} \left| p_{\widehat{Y},S|Y}(\widehat{y}|s,Y) - p_{\widehat{Y}|Y}(\widehat{y}|Y) \right| \middle| Y \in \mathcal{A} \right\}. \tag{29}$$

**Lemma 9** (ERMI provides an upper bound for conditional equal opportunity $L_\infty$ violation)**.** *Let $\widehat{Y}, Y$, be discrete or continuous random variables supported on $\mathcal{Y}$, and let $S$ be a discrete random variable supported on a finite set $\mathcal{S}$. Let $\mathcal{A} \subseteq \mathcal{Y}$ be a compact subset of $\mathcal{Y}$.*

*Denote $p_{S|\mathcal{A}}^{\min} = \min_{s\in\mathcal{S}, y\in\mathcal{A}} p_{S|Y}(s|y)$. Then,*

$$0 \leq \widetilde{eo}(\widehat{Y}|S, Y \in \mathcal{A}) \leq \frac{1}{p_{S|\mathcal{A}}^{\min}}\sqrt{D_R(\widehat{Y};S|Y \in \mathcal{A})}. \tag{30}$$

*Proof.* Notice that the same proof for Lemma 8 would give that for all $y \in \mathcal{A}$:

$$0 \leq \sup_{\widehat{y}\in\mathcal{Y}} \max_{s\in\mathcal{S}} \left| p_{\widehat{Y},S|Y}(\widehat{y}|s,y) - p_{\widehat{Y}|Y}(\widehat{y}|y) \right| := \widetilde{eo}(\widehat{Y}|S, Y = y)$$

$$\leq \frac{1}{p_{S|y}^{\min}(y)}\sqrt{D_R(\widehat{Y};S|Y=y)}$$

$$\leq \frac{1}{p_{S|\mathcal{C}}^{\min}}\sqrt{D_R(\widehat{Y};S|Y=y)}.$$

Hence,

$$\widetilde{\text{eo}}(\widehat{Y}|S, Y \in \mathcal{A}) = \mathbb{E}_Y \left\{ \widetilde{\text{eo}}(\widehat{Y}|S,Y) \,\middle|\, Y \in \mathcal{A} \right\}$$

$$\leq \frac{1}{p_{S|\mathcal{A}}^{\min}} \mathbb{E}_Y \left\{ \sqrt{D_R(\widehat{Y}; S|Y)} \,\middle|\, Y \in \mathcal{A} \right\}$$

$$\leq \frac{1}{p_{S|\mathcal{A}}^{\min}} \sqrt{\mathbb{E}_Y \left\{ D_R(\widehat{Y}; S|Y) \,\middle|\, Y \in \mathcal{A} \right\}}$$

$$= \frac{1}{p_{S|\mathcal{A}}^{\min}} \sqrt{D_R(\widehat{Y}; S|Y \in \mathcal{A})},$$

where the last inequality follows from Jensen's inequality. This completes the proof. $\square$

## D  Precise Statement and Proofs of Theorem 1 and Theorem 2

To begin, we provide the proof of Proposition 1:

**Proposition 3** (Re-statement of Proposition 1). *For random variables $\widehat{Y}$ and $S$ with joint distribution $\hat{p}_{\widehat{Y},S}$, where $\widehat{Y} \in [m], S \in [k]$, we have*

$$\widehat{D}_R(\widehat{Y}; S) = \max_{W \in \mathbb{R}^{k \times m}} \left\{ -\operatorname{Tr}(W\widehat{P}_{\hat{y}}W^T) + 2\operatorname{Tr}(W\widehat{P}_{\hat{y},s}\widehat{P}_s^{-1/2}) - 1 \right\},$$

*if $\widehat{P}_{\hat{y}} = \operatorname{diag}(\hat{p}_{\widehat{Y}}(1), \ldots, \hat{p}_{\widehat{Y}}(m))$, $\widehat{P}_s = \operatorname{diag}(\hat{p}_S(1), \ldots, \hat{p}_S(k))$, and $(\widehat{P}_{\hat{y},s})_{i,j} = \hat{p}_{\widehat{Y},S}(i,j)$ with $\hat{p}_{\widehat{Y}}(i), \hat{p}_S(j) > 0$ for $i \in [m], j \in [k]$.*

*Proof.* Let $W^* \in \arg\max_{W \in \mathbb{R}^{k \times m}} -\operatorname{Tr}(W\widehat{P}_{\hat{y}}W^T) + 2\operatorname{Tr}(W\widehat{P}_{\hat{y},s}\widehat{P}_s^{-1/2})$. Setting the derivative of the expression on the RHS equal to zero leads to:

$$-2W\widehat{P}_{\hat{y}} + 2\widehat{P}_s^{-1/2}\widehat{P}_{\hat{y},s}^T = 0 \implies W^* = \widehat{P}_s^{-1/2}\widehat{P}_{\hat{y},s}^T\widehat{P}_{\hat{y}}^{-1}.$$

Plugging this expression for $W^*$, we have

$$
\begin{aligned}
\max_{W \in \mathbb{R}^{k \times m}} & -\operatorname{Tr}(W\widehat{P}_{\hat{y}}W^T) + 2\operatorname{Tr}(W\widehat{P}_{\hat{y},s}\widehat{P}_s^{-1/2}) \\
&= -\operatorname{Tr}(\widehat{P}_s^{-1/2}\widehat{P}_{\hat{y},s}^T P_{\hat{y}}^{-1}\widehat{P}_{\hat{y},s}\widehat{P}_s^{-1/2}) + 2\operatorname{Tr}(\widehat{P}_s^{-1/2}\widehat{P}_{\hat{y},s}^T P_{\hat{y}}^{-1}\widehat{P}_{\hat{y},s}\widehat{P}_s^{-1/2}) \\
&= \operatorname{Tr}(\widehat{P}_s^{-1/2}\widehat{P}_{\hat{y},s}^T P_{\hat{y}}^{-1}\widehat{P}_{\hat{y},s}\widehat{P}_s^{-1/2}) \\
&= \operatorname{Tr}(\widehat{P}_s^{-1}\widehat{P}_{\hat{y},s}^T P_{\hat{y}}^{-1}\widehat{P}_{\hat{y},s}).
\end{aligned}
$$

Writing out the matrix multiplication explicitly in the last expression, we have

$$\widehat{P}_s^{-1}\widehat{P}_{\hat{y},s}^T \widehat{P}_{\hat{y}}^{-1}\widehat{P}_{\hat{y},s} = UV^T,$$

where $U_{i,j} = \hat{p}_S(i)^{-1}\hat{p}_{\widehat{Y},S}(j,i)$ and $V_{i,j} = \hat{p}_{\widehat{Y}}(j)^{-1}\hat{p}_{\widehat{Y},S}(j,i)$, for $i \in [k], j \in [m]$. Hence

$$
\begin{aligned}
\max_{W \in \mathbb{R}^{k \times m}} & -\operatorname{Tr}(W\widehat{P}_{\hat{y}}W^T) + 2\operatorname{Tr}(W\widehat{P}_{\hat{y},s}\widehat{P}_s^{-1/2}) \\
&= \operatorname{Tr}(UV^T) \\
&= \sum_{i \in [k]} \sum_{j \in [m]} \frac{\hat{p}_{\widehat{Y},S}(j,i)^2}{\hat{p}_S(i)\hat{p}_{\widehat{Y}}(j)} \\
&= \widehat{D}_R(\widehat{Y}; S) + 1,
\end{aligned}
$$

which completes the proof. $\qquad\square$

**Corollary 2** (Re-statement of Corollary 1). *Let $(\mathbf{x}_i, s_i, y_i)$ be a random draw from $\mathbf{D}$. Then, Eq.* (FERMI obj.) *is equivalent to*

$$\min_{\boldsymbol{\theta}} \max_{W \in \mathbb{R}^{k \times m}} \left\{ \widehat{F}(\boldsymbol{\theta}, W) := \widehat{\mathcal{L}}(\boldsymbol{\theta}) + \lambda\widehat{\Psi}(\boldsymbol{\theta}, W) \right\}, \tag{31}$$

*where $\widehat{\Psi}(\boldsymbol{\theta}, W) = -\operatorname{Tr}(W\widehat{P}_{\hat{y}}W^T) + 2\operatorname{Tr}(W\widehat{P}_{\hat{y},s}\widehat{P}_s^{-1/2}) - 1 = \frac{1}{N}\sum_{i=1}^N \widehat{\psi}_i(\boldsymbol{\theta}, W)$ and*

$$
\begin{aligned}
\widehat{\psi}_i(\boldsymbol{\theta}, W) &:= -\operatorname{Tr}(W\mathbb{E}[\widehat{\mathbf{y}}(\mathbf{x}_i, \boldsymbol{\theta})\widehat{\mathbf{y}}(\mathbf{x}_i, \boldsymbol{\theta})^T|\mathbf{x}_i]W^T) + 2\operatorname{Tr}(W\mathbb{E}[\widehat{\mathbf{y}}(\mathbf{x}_i; \boldsymbol{\theta})\mathbf{s}_i^T|\mathbf{x}_i, s_i]\widehat{P}_s^{-1/2}) - 1 \\
&= -\operatorname{Tr}(W\operatorname{diag}(\mathcal{F}_1(\boldsymbol{\theta}, \mathbf{x}_i), \ldots, \mathcal{F}_m(\boldsymbol{\theta}, \mathbf{x}_i))W^T) + 2\operatorname{Tr}(W\mathbb{E}[\widehat{\mathbf{y}}(\mathbf{x}_i; \boldsymbol{\theta})\mathbf{s}_i^T|\mathbf{x}_i, s_i]\widehat{P}_s^{-1/2}) - 1.
\end{aligned}
$$

*Proof.* The proof simply follows the fact that

$$\max_{W \in \mathbb{R}^{k \times m}} \mathbb{E}\left[\widehat{\psi}_i(\boldsymbol{\theta}, W)\right] = \max_{W \in \mathbb{R}^{k \times m}} \left( -\operatorname{Tr}(W\widehat{P}_{\hat{y}}W^T) + 2\operatorname{Tr}(W\widehat{P}_{\hat{y},s}\widehat{P}_s^{-1/2}) - 1 \right) = \widehat{D}_R(\widehat{Y}; S),$$

where the last equality is due to Proposition 1. $\qquad\square$

Next, we will state and prove the precise form of Theorem 1. We first recall some basic definitions:

**Definition 12.** *A function $f$ is $L$-Lipschitz if for all $\mathbf{u}, \mathbf{u}' \in domain(f)$ we have $\|f(\mathbf{u}) - f(\mathbf{u})\| \leq L\|\mathbf{u} - \mathbf{u}'\|$.*

**Definition 13.** *A differentiable function $f$ is $\beta$-smooth if for all $\mathbf{u}, \mathbf{u}' \in domain(\nabla f)$ we have $\|\nabla f(\mathbf{u}) - \nabla f(\mathbf{u})\| \leq \beta\|\mathbf{u} - \mathbf{u}'\|$.*

**Definition 14.** *A differentiable function $f$ is $\mu$-strongly concave if for all $\mathbf{x}, \mathbf{y} \in domain(f)$, we have $f(\mathbf{x}) + f(\mathbf{x})^T(\mathbf{y} - \mathbf{x}) - \frac{\mu}{2}\|\mathbf{y} - \mathbf{x}\|^2 \geq f(\mathbf{y})$*

**Definition 15.** *A point $\boldsymbol{\theta}^* = \mathcal{A}(\mathbf{D})$ output by a randomized algorithm $\mathcal{A}$ is an $\epsilon$-stationary point of a differentiable function $\Phi$ if $\mathbb{E}\|\nabla\Phi(\boldsymbol{\theta}^*)\| \leq \epsilon$. We say $\boldsymbol{\theta}^*$ is an $\epsilon$-stationary point of the nonconvex-strongly concave min-max problem $\min_{\boldsymbol{\theta}} \max_W F(\boldsymbol{\theta}, W)$ if it is an $\epsilon$-stationary point of the differentiable function $\Phi(\boldsymbol{\theta}) := \max_W F(\boldsymbol{\theta}, W)$.*

Recall that Eq. (FERMI obj.) is equivalent to

$$\min_{\boldsymbol{\theta}} \max_{W \in \mathbb{R}^{k \times m}} \left\{ \widehat{F}(\boldsymbol{\theta}, W) := \widehat{\mathcal{L}}(\boldsymbol{\theta}) + \lambda\widehat{\Psi}(\boldsymbol{\theta}, W) = \frac{1}{N}\sum_{i=1}^{N}\left[ \ell(\mathbf{x}_i, y_i, \boldsymbol{\theta}) + \lambda\widehat{\psi}_i(\boldsymbol{\theta}, W) \right] \right\}, \quad (32)$$

where $\widehat{\Psi}(\boldsymbol{\theta}, W) = -\operatorname{Tr}(W\widehat{P}_{\hat{y}}W^T) + 2\operatorname{Tr}(W\widehat{P}_{\hat{y},s}\widehat{P}_s^{-1/2}) - 1 = \frac{1}{N}\sum_{i=1}^{N}\widehat{\psi}_i(\boldsymbol{\theta}, W)$ and

$$\widehat{\psi}_i(\boldsymbol{\theta}, W) := -\operatorname{Tr}(W\mathbb{E}[\widehat{\mathbf{y}}(\mathbf{x}_i, \boldsymbol{\theta})\widehat{\mathbf{y}}(\mathbf{x}_i, \boldsymbol{\theta})^T | \mathbf{x}_i]W^T) + 2\operatorname{Tr}(W\mathbb{E}[\widehat{\mathbf{y}}(\mathbf{x}_i; \boldsymbol{\theta})\mathbf{s}_i^T | \mathbf{x}_i, s_i]\widehat{P}_s^{-1/2}) - 1$$
$$= -\operatorname{Tr}(W\operatorname{diag}(\mathcal{F}_1(\boldsymbol{\theta}, \mathbf{x}_i), \ldots, \mathcal{F}_m(\boldsymbol{\theta}, \mathbf{x}_i))W^T) + 2\operatorname{Tr}(W\mathbb{E}[\widehat{\mathbf{y}}(\mathbf{x}_i; \boldsymbol{\theta})\mathbf{s}_i^T | \mathbf{x}_i, s_i]\widehat{P}_s^{-1/2}) - 1,$$

where $\widehat{\mathbf{y}}(\mathbf{x}_i; \boldsymbol{\theta})$ and $\mathbf{s}_i$ are the one-hot encodings of $\widehat{y}(\mathbf{x}_i; \boldsymbol{\theta})$ and $s_i$, respectively.

**Assumption 1.**
- $\ell(\cdot, \mathbf{x}, y)$ *is $G$-Lipscthiz, and $\beta_\ell$-smooth for all $\mathbf{x}, y$.*

- $\mathcal{F}(\cdot, \mathbf{x})$ *is $L$-Lipschitz and $b$-smooth for all $\mathbf{x}$.*

- $\widehat{p}_{\hat{y}}^{\min} := \inf_{\{\boldsymbol{\theta}^t, t \in [T]\}} \min_{j \in [m]} \frac{1}{N}\sum_{i=1}^{N} \mathcal{F}_j(\boldsymbol{\theta}, x_i) \geq \frac{\mu}{2} > 0.$

- $\hat{p}_S^{\min} := \frac{1}{N}\sum_{i=1}^{N} \mathbb{1}_{\{s_i = j\}} > 0.$

**Remark 3.** *As mentioned in remark 1, the third bullet in Assumption 1 is convenient and allows for a faster convergence rate, but not strictly necessary for convergence of Algorithm 1.*

**Theorem 3** (Precise statement of Theorem 1). *Let $\{\mathbf{x}_i, \mathbf{y}_i, \mathbf{s}_i\}_{i \in [N]}$ be any given data set of features, labels, and sensitive attributes and grant Assumption 1. Let $\mathcal{W} := B_F(0, D) \subset \mathbb{R}^{k \times m}$ (Frobenius norm ball of radius $D$), where $D := \frac{2}{\hat{p}_{\hat{Y}}^{\min}\sqrt{\hat{p}_S^{\min}}}$ in Algorithm 1. Denote $\Delta_{\widehat{\Phi}} := \widehat{\Phi}(\boldsymbol{\theta}_0) - \inf_{\boldsymbol{\theta}}\widehat{\Phi}(\theta)$, where $\widehat{\Phi}(\boldsymbol{\theta}) := \max_W \widehat{F}(\boldsymbol{\theta}, W)$. In Algorithm 1, choose the step-sizes as $\eta_\theta = \Theta(1/\kappa^2\beta)$ and $\eta_W = \Theta(1/\beta)$ and mini-batch size as $|B_t| = \Theta\left(\max\{1, \kappa\sigma^2\epsilon^{-2}\}\right)$. Then under Assumption 1, the iteration complexity of Algorithm 1 to return an $\epsilon$-stationary point of $\widehat{\Phi}$ is bounded by*

$$\mathcal{O}\left(\frac{\kappa^2\beta\Delta_{\widehat{\Phi}} + \kappa\beta^2 D^2}{\epsilon^2}\right),$$

*which gives the total stochastic gradient complexity of*

$$\mathcal{O}\left(\left(\frac{\kappa^2\beta\Delta_{\widehat{\Phi}} + \kappa\beta^2 D^2}{\epsilon^2}\right)\max\{1, \kappa\sigma^2\epsilon^{-2}\}\right),$$

*where*

$$\beta = 2\left(\beta_\ell + 2\lambda Dmb\left(D + \frac{1}{\sqrt{\hat{p}_S^{\min}}}\right) + 2 + 8L\left(D + \frac{1}{\sqrt{\hat{p}_S^{\min}}}\right)\right),$$
$$\mu = 2\lambda\hat{p}_{\hat{Y}}^{\min},$$
$$\kappa = \beta/\mu,$$
$$\sigma^2 = 16\lambda^2(D^2 + 1) + 4G^2 + 32\lambda^2 D^2 L^2\left(1 + \frac{mk}{\hat{p}_S^{\min}}\right).$$

**Remark 4.** *The larger minibatch size is necessary to obtain the faster $\mathcal{O}(\epsilon^{-4})$ convergence rate via two-timescale SGDA. However, as noted in [Lin et al., 2020, p.8], their proof readily extends to any batch size $|B_t| \geq 1$, showing that two-timescale SGDA still converges. But with $|B_t| = 1$, the iteration complexity becomes slower: $\mathcal{O}(\kappa^3 \epsilon^{-5})$. This is the informal Theorem 1 that was stated in the main body.*

In light of Corollary 1, Theorem 3 follows from [Lin et al., 2020, Theorem 4.5] combined with the following technical lemmas. We assume Assumption 1 holds for the remainder of the proof of Theorem 3:

**Lemma 10.** *If $\mathbf{x}_i, y_i, s_i$ are drawn uniformly at random from data set $\mathbf{D}$, then the gradients of $\ell(\mathbf{x}_i, y_i, \boldsymbol{\theta}) + \lambda \nabla \widehat{\psi}_i(\boldsymbol{\theta}, W)$ are unbiased estimators of the gradients of $\widehat{F}(\boldsymbol{\theta}, W)$ for all $\boldsymbol{\theta}, W, \lambda$:*

$$\mathbb{E}[\nabla_{\boldsymbol{\theta}} \ell(\mathbf{x}_i, y_i, \boldsymbol{\theta}) + \lambda \nabla_{\boldsymbol{\theta}} \widehat{\psi}_i(\boldsymbol{\theta}, W)] = \nabla_{\boldsymbol{\theta}} \widehat{F}(\boldsymbol{\theta}, W), \text{ and}$$

$$\mathbb{E}[\lambda \nabla_W \widehat{\psi}_i(\boldsymbol{\theta}, W)] = \nabla_W \widehat{F}(\boldsymbol{\theta}, W).$$

*Furthermore, if $\|W\|_F \leq D$, then the variance of the stochastic gradients is bounded as follows:*

$$\sup_{\boldsymbol{\theta}, W} \mathbb{E}\|\nabla \ell(\mathbf{x}_i, y_i, \boldsymbol{\theta}) + \lambda \nabla \widehat{\psi}_i(\boldsymbol{\theta}, W) - \nabla \widehat{F}(\boldsymbol{\theta}, W)\|^2 \leq \sigma^2, \tag{33}$$

*where $\sigma^2 = 16\lambda^2(D^2 + 1) + 4G^2 + 32\lambda^2 D^2 L^2 \left(1 + \frac{mk}{\widehat{p}_S^{\min}}\right)$.*

*Proof.* Unbiasedness is obvious. For the variance bound, we will show that

$$\sup_{\boldsymbol{\theta}, W} \mathbb{E}\|\lambda \nabla_W \widehat{\psi}_i(\boldsymbol{\theta}, W) - \nabla_W \widehat{F}(\boldsymbol{\theta}, W)\|^2 \leq \sigma_w^2, \tag{34}$$

and

$$\sup_{\boldsymbol{\theta}, W} \mathbb{E}\|\nabla_{\boldsymbol{\theta}} \ell(\mathbf{x}_i, y_i, \boldsymbol{\theta}) + \lambda \nabla_{\boldsymbol{\theta}} \widehat{\psi}_i(\boldsymbol{\theta}, W) - \nabla_{\boldsymbol{\theta}} \widehat{F}(\boldsymbol{\theta}, W)\|^2 \leq \sigma_{\boldsymbol{\theta}}^2, \tag{35}$$

where $\sigma^2 = \sigma_{\boldsymbol{\theta}}^2 + \sigma_w^2$. First,

$$\nabla_W \widehat{\psi}_i(\boldsymbol{\theta}, W) = -2W \mathbb{E}[\widehat{\mathbf{y}}(\mathbf{x}_i, \boldsymbol{\theta})\widehat{\mathbf{y}}(\mathbf{x}_i, \boldsymbol{\theta})^T | \mathbf{x}_i] + 2\widehat{p}_s(r)^{-1/2} \mathbb{E}[\mathbf{s}_i \widehat{\mathbf{y}}(\mathbf{x}_i, \boldsymbol{\theta}) | \mathbf{x}_i, s_i]. \tag{36}$$

Thus, for any $\boldsymbol{\theta}, W, \lambda$, we have

$$
\begin{aligned}
\mathbb{E}\|\lambda \nabla_W \widehat{\psi}_i(\boldsymbol{\theta}, W) - \nabla_W \widehat{F}(\boldsymbol{\theta}, W)\|_F^2 = {} & \frac{4\lambda^2}{N} \sum_{i=1}^N \Bigg\| W \mathbb{E}[\widehat{\mathbf{y}}(\mathbf{x}_i, \boldsymbol{\theta})\widehat{\mathbf{y}}(\mathbf{x}_i, \boldsymbol{\theta})^T | \mathbf{x}_i] - \widehat{p}_s(r)^{-1/2} \mathbb{E}[\mathbf{s}_i \widehat{\mathbf{y}}(\mathbf{x}_i, \boldsymbol{\theta})^T | \mathbf{x}_i, s_i] \\
& - \frac{1}{N} \sum_{i=1}^N \left( W \mathbb{E}[\widehat{\mathbf{y}}(\mathbf{x}_i, \boldsymbol{\theta})\widehat{\mathbf{y}}(\mathbf{x}_i, \boldsymbol{\theta})^T | \mathbf{x}_i] - \widehat{p}_s(r)^{-1/2} \mathbb{E}[\mathbf{s}_i \widehat{\mathbf{y}}(\mathbf{x}_i, \boldsymbol{\theta})^T | \mathbf{x}_i, s_i] \right) \Bigg\|_F^2 \\
\leq {} & \frac{4\lambda^2}{N} \sum_{i=1}^N 2 \Bigg[ \|W\|_F^2 \left\| \mathbb{E}[\widehat{\mathbf{y}}(\mathbf{x}_i, \boldsymbol{\theta})\widehat{\mathbf{y}}(\mathbf{x}_i, \boldsymbol{\theta})^T | \mathbf{x}_i] - \widehat{P}_{\widehat{y}} \right\|_F^2 \\
& + \left\| \widehat{P}_s^{-1/2} \left( \mathbb{E}[\mathbf{s}_i \widehat{\mathbf{y}}(\mathbf{x}_i, \boldsymbol{\theta})^T | \mathbf{x}_i, s_i] - \widehat{P}_{\widehat{y}, s}^T \right) \right\|_F^2 \Bigg] \\
\leq {} & \frac{4\lambda^2}{N} \sum_{i=1}^N 2 \left[ 2D^2 + \left\| \widehat{P}_s^{-1/2} \left( \mathbb{E}[\mathbf{s}_i \widehat{\mathbf{y}}(\mathbf{x}_i, \boldsymbol{\theta})^T | \mathbf{x}_i, s_i] - \widehat{P}_{\widehat{y}, s}^T \right) \right\|_F^2 \right] \\
\leq {} & \frac{4\lambda^2}{N} \sum_{i=1}^N 2 \left[ 2D^2 + 2 \right] \\
\leq {} & 16\lambda^2(D^2 + 1),
\end{aligned}
$$

where we used Young's inequality, the Frobenius norm inequality $\|AB\|_F \leq \|A\|_F \|B\|_F$, the facts that $\|\mathbb{E}[\widehat{\mathbf{y}}(\mathbf{x}_i, \boldsymbol{\theta})\widehat{\mathbf{y}}(\mathbf{x}_i, \boldsymbol{\theta})^T | \mathbf{x}_i]\|_F^2 = \sum_{j=1}^m \mathcal{F}_j(\boldsymbol{\theta}, \mathbf{x}_i)^2 \leq 1$ and $\|\widehat{P}_s^{-1/2} \mathbb{E}[\mathbf{s}_i \widehat{\mathbf{y}}(\mathbf{x}_i, \boldsymbol{\theta})^T | \mathbf{x}_i, s_i]\|_F^2 = \sum_{j=1}^k \mathbf{s}_{i,j}^2 \sum_{l=1}^m \mathcal{F}_l(\mathbf{x}_i, \boldsymbol{\theta})^2 \leq 1$ for all $i \in [N]$ (since for every $i \in [N]$, only one of the $\mathbf{s}_{i,j}$ is

non-zero and equal to 1, and $\sum_{l=1}^{m} \mathcal{F}_l(\boldsymbol{\theta}, \mathbf{x}_i) = 1$).

Next,

$$\lambda \nabla_{\boldsymbol{\theta}} \widehat{\psi}_i(\boldsymbol{\theta}, W) = \lambda \left[ -\nabla_{\boldsymbol{\theta}} \operatorname{vec}(\mathbb{E}[\widehat{\mathbf{y}}(\mathbf{x}_i, \boldsymbol{\theta}) \widehat{\mathbf{y}}(\mathbf{x}_i, \boldsymbol{\theta})^T | \mathbf{x}_i])^T \operatorname{vec}(W^T W) + 2 \nabla_{\boldsymbol{\theta}} \operatorname{vec}(\mathbb{E}[\mathbf{s}_i \widehat{\mathbf{y}}(\mathbf{x}_i, \boldsymbol{\theta})^T | \mathbf{x}_i, s_i]) \operatorname{vec}(W^T \widehat{P}_s^{-1/2}) \right]. \tag{37}$$

Hence, for any $\boldsymbol{\theta}, W$, we have

$$\mathbb{E}\|\nabla_{\boldsymbol{\theta}} \ell(\mathbf{x}_i, y_i, \boldsymbol{\theta}) + \lambda \nabla_{\boldsymbol{\theta}} \widehat{\psi}_i(\boldsymbol{\theta}, W) - \nabla_{\boldsymbol{\theta}} \widehat{F}(\boldsymbol{\theta}, W)\|^2 \le 2 \left[ 2 \sup_{\mathbf{x}_i, y_i} \|\nabla_{\boldsymbol{\theta}} \ell(\mathbf{x}_i, y_i, \boldsymbol{\theta})\|^2 \right.$$

$$+ \lambda^2 \sup_{\mathbf{x}_i, y_i, s_i} \left\| -\nabla_{\boldsymbol{\theta}} \operatorname{vec}(\mathbb{E}[\widehat{\mathbf{y}}(\mathbf{x}_i, \boldsymbol{\theta}) \widehat{\mathbf{y}}(\mathbf{x}_i, \boldsymbol{\theta})^T | \mathbf{x}_i])^T \operatorname{vec}(W^T W) \right.$$

$$\left. \left. + 2 \nabla_{\boldsymbol{\theta}} \operatorname{vec}(\mathbb{E}[\mathbf{s}_i \widehat{\mathbf{y}}(\mathbf{x}_i, \boldsymbol{\theta})^T | \mathbf{x}_i, s_i]) \operatorname{vec}(W^T \widehat{P}_s^{-1/2}) \right\|^2 \right]$$

$$\le 4 \left[ G^2 + 2\lambda^2 \left( \sup_{\mathbf{x}_i} \left\| \nabla_{\boldsymbol{\theta}} \operatorname{vec}(\mathbb{E}[\widehat{\mathbf{y}}(\mathbf{x}_i, \boldsymbol{\theta}) \widehat{\mathbf{y}}(\mathbf{x}_i, \boldsymbol{\theta})^T | \mathbf{x}_i])^T \operatorname{vec}(W^T W) \right\| \right. \right.$$

$$\left. \left. + 4 \sup_{\mathbf{x}_i, s_i} \left\| \nabla_{\boldsymbol{\theta}} \operatorname{vec}(\mathbb{E}[\mathbf{s}_i \widehat{\mathbf{y}}(\mathbf{x}_i, \boldsymbol{\theta})^T | \mathbf{x}_i, s_i]) \operatorname{vec}(W^T \widehat{P}_s^{-1/2}) \right\|^2 \right) \right],$$

by Young's and Jensen's inequalities and the assumption that $\ell(\mathbf{x}_i, y_i, \cdot)$ is $G$-Lipschitz. Now,

$$\nabla_{\boldsymbol{\theta}} \operatorname{vec}(\mathbb{E}[\widehat{\mathbf{y}}(\mathbf{x}_i, \boldsymbol{\theta}) \widehat{\mathbf{y}}(\mathbf{x}_i, \boldsymbol{\theta})^T | \mathbf{x}_i])^T \operatorname{vec}(W^T W) = \sum_{l=1}^{m} \nabla \mathcal{F}_l(\mathbf{x}_i, \boldsymbol{\theta}) \sum_{j=1}^{k} W_{j,1} W_{j,l},$$

which implies

$$\|\nabla_{\boldsymbol{\theta}} \operatorname{vec}(\mathbb{E}[\widehat{\mathbf{y}}(\mathbf{x}_i, \boldsymbol{\theta}) \widehat{\mathbf{y}}(\mathbf{x}_i, \boldsymbol{\theta})^T | \mathbf{x}_i])^T \operatorname{vec}(W^T W)\|^2 \le \sum_{j,l} W_{j,l}^2 \sup_{l \in [m], \mathbf{x}, \boldsymbol{\theta}} \|\nabla \mathcal{F}_l(\mathbf{x}, \boldsymbol{\theta})\|^2 \le D^2 L^2, \tag{38}$$

by $L$-Lipschitzness of $\mathcal{F}(\cdot, \mathbf{x})$. Also,

$$\nabla_{\boldsymbol{\theta}} \operatorname{vec}(\mathbb{E}[\mathbf{s}_i \widehat{\mathbf{y}}(\mathbf{x}_i, \boldsymbol{\theta})^T | \mathbf{x}_i, s_i]) \operatorname{vec}(W^T \widehat{P}_s^{-1/2}) = \sum_{r=1}^{k} \sum_{j=1}^{m} \nabla \mathcal{F}_j(\boldsymbol{\theta}, \mathbf{x}_i) \frac{\mathbf{s}_{i,r} W_{r,j}}{\sqrt{\hat{p}_S(r)}},$$

which implies

$$\left\| \nabla_{\boldsymbol{\theta}} \operatorname{vec}(\mathbb{E}[\mathbf{s}_i \widehat{\mathbf{y}}(\mathbf{x}_i, \boldsymbol{\theta})^T | \mathbf{x}_i, s_i]) \operatorname{vec}(W^T \widehat{P}_s^{-1/2}) \right\|^2 \le mk \sum_{r=1}^{k} \sum_{j=1}^{m} \sup_{\mathbf{x}_i, \boldsymbol{\theta}} \|\nabla \mathcal{F}_j(\boldsymbol{\theta}, \mathbf{x}_i)\|^2 \left( \frac{\mathbf{s}_{i,r} W_{r,j}}{\sqrt{\hat{p}_S(r)}} \right)^2 \le \frac{mk}{\hat{p}_S^{\min}} L^2 D^2.$$

Thus,

$$\sigma_{\boldsymbol{\theta}}^2 \le 4G^2 + 32\lambda^2 D^2 L^2 \left( 1 + \frac{mk}{\hat{p}_S^{\min}} \right).$$

Combining the $\boldsymbol{\theta}$- and $W$-variance bounds yields the lemma. $\qquad \square$

**Lemma 11.** *Let*

$$\widehat{F}(\boldsymbol{\theta}, W) = \frac{1}{N} \sum_{i \in [N]} \ell(\mathbf{x}_i, y_i; \boldsymbol{\theta}) + \lambda \widehat{\psi}_i(\boldsymbol{\theta}, W)$$

*where*

$$\widehat{\psi}_i(\boldsymbol{\theta}, W) = -\operatorname{Tr}(W \mathbb{E}[\widehat{\mathbf{y}}(\mathbf{x}_i, \boldsymbol{\theta}) \widehat{\mathbf{y}}(\mathbf{x}_i, \boldsymbol{\theta})^T | \mathbf{x}_i] W^T) + 2 \operatorname{Tr}(W \mathbb{E}[\widehat{\mathbf{y}}(\mathbf{x}_i; \boldsymbol{\theta}) \mathbf{s}_i^T | \mathbf{x}_i, s_i] \widehat{P}_s^{-1/2}) - 1.$$

*Then:*

1. *$\widehat{F}$ is $\beta$-smooth, where $\beta = 2 \left( \beta_\ell + 2\lambda Dmb \left( D + \frac{1}{\sqrt{\hat{p}_S^{\min}}} \right) + 2 + 8L \left( D + \frac{1}{\sqrt{\hat{p}_S^{\min}}} \right) \right)$.*

2. $\widehat{F}(\boldsymbol{\theta}, \cdot)$ is $2\lambda \hat{p}_{\widehat{Y}}^{\min}$-strongly concave for all $\boldsymbol{\theta}^t$.

3. If $\mathcal{W} = B_F(0, D)$ with $D \geq \frac{2}{\hat{p}_{\widehat{Y}}^{\min} \sqrt{\hat{p}_S^{\min}}}$, then Eq. (1) $= \min_{\boldsymbol{\theta}} \max_{W \in \mathcal{W}} \widehat{F}(\boldsymbol{\theta}, W)$.

*Proof.* We shall freely use the expressions for the derivatives of $\widehat{\psi}_i$ obtained in the proof of Lemma 10.
1. First,

$$\|\nabla_w \widehat{F}(\boldsymbol{\theta}, W) - \nabla_w \widehat{F}(\boldsymbol{\theta}, W')\|_F \leq 2 \sup_{\mathbf{x}_i} \|W \mathbb{E}[\widehat{\mathbf{y}}(\mathbf{x}_i, \boldsymbol{\theta})\widehat{\mathbf{y}}(\mathbf{x}_i, \boldsymbol{\theta})^T | \mathbf{x}_i] - W' \mathbb{E}[\widehat{\mathbf{y}}(\mathbf{x}_i, \boldsymbol{\theta})\widehat{\mathbf{y}}(\mathbf{x}_i, \boldsymbol{\theta})^T | \mathbf{x}_i]\|_F$$

$$\leq 2\|W - W'\|_F,$$

since $\mathcal{F}_j(\boldsymbol{\theta}, \mathbf{x}_i) \leq 1$ for all $j \in [m]$. Next,

$$\|\nabla_w \widehat{F}(\boldsymbol{\theta}, W) - \nabla_w \widehat{F}(\boldsymbol{\theta}', W)\|_F^2$$

$$\leq 8 \sup_{\mathbf{x}_i, \mathbf{s}_i, y_i} \left[ D^2 \left\| \mathbb{E}[\widehat{\mathbf{y}}(\mathbf{x}_i, \boldsymbol{\theta})\widehat{\mathbf{y}}(\mathbf{x}_i, \boldsymbol{\theta})^T | \mathbf{x}_i] - \mathbb{E}[\widehat{\mathbf{Y}}(\mathbf{x}_i, \boldsymbol{\theta}')\widehat{\mathbf{Y}}(\mathbf{x}_i, \boldsymbol{\theta}')^T | \mathbf{x}_i] \right\|_F^2 \right.$$

$$\left. + \left\| \widehat{P}_s^{-1/2} \left( \mathbb{E}[\mathbf{s}_i \widehat{\mathbf{y}}(\mathbf{x}_i, \boldsymbol{\theta})^T | \mathbf{x}_i, s_i] - \mathbb{E}[\mathbf{s}_i \widehat{\mathbf{Y}}(\mathbf{x}_i, \boldsymbol{\theta}')^T | \mathbf{x}_i, s_i] \right) \right\|_F^2 \right]$$

$$\leq 8 \sup_{\mathbf{x}_i, \mathbf{s}_i, y_i} \left[ D^2 \|\mathcal{F}(\boldsymbol{\theta}, \mathbf{x}_i) - \mathcal{F}(\boldsymbol{\theta}', \mathbf{x}_i)\|_F^2 \right.$$

$$\left. + \sum_{j=1}^m \sum_{r=1}^k |\mathcal{F}_j(\boldsymbol{\theta}, \mathbf{x}_i) - \mathcal{F}_j(\boldsymbol{\theta}', \mathbf{x}_i)|^2 \widehat{p}_s(r)(r)^{-1} \mathbf{s}_{i,r}^2 \right]$$

$$\leq 8 \sup_{\mathbf{x}_i, \mathbf{s}_i, y_i} \left[ D^2 L^2 \|\boldsymbol{\theta} - \boldsymbol{\theta}'\|^2 + \frac{L^2}{\hat{p}_S^{\min}} \|\boldsymbol{\theta} - \boldsymbol{\theta}'\|^2 \right],$$

which implies

$$\|\nabla_w \widehat{F}(\boldsymbol{\theta}, W) - \nabla_w \widehat{F}(\boldsymbol{\theta}', W)\|_F \leq 8L \left( D + \frac{1}{\sqrt{\hat{p}_S^{\min}}} \right) \|\boldsymbol{\theta} - \boldsymbol{\theta}'\|.$$

Lastly,

$$\|\nabla_{\boldsymbol{\theta}} \widehat{F}(\boldsymbol{\theta}, W) - \nabla_{\boldsymbol{\theta}} \widehat{F}(\boldsymbol{\theta}', W)\| \leq \sup_{\mathbf{x}_i, y_i, s_i} \left[ \|\nabla \ell(\mathbf{x}_i, y_i, \boldsymbol{\theta}) - \nabla \ell(\mathbf{x}_i, y_i, \boldsymbol{\theta}')\| \right.$$

$$+ \lambda \left\| \left[ -\nabla_{\boldsymbol{\theta}} \text{vec}(\mathbb{E}[\widehat{\mathbf{y}}(\mathbf{x}_i, \boldsymbol{\theta})\widehat{\mathbf{y}}(\mathbf{x}_i, \boldsymbol{\theta})^T | \mathbf{x}_i])^T + \nabla_{\boldsymbol{\theta}} \text{vec}(\mathbb{E}[\widehat{\mathbf{y}}(\mathbf{x}_i, \boldsymbol{\theta}')\widehat{\mathbf{y}}(\mathbf{x}_i, \boldsymbol{\theta}')^T | \mathbf{x}_i])^T \right] \right.$$

$$\cdot \text{vec}(W^T W) \right\| + 2\lambda \left\| [\nabla_{\boldsymbol{\theta}} \text{vec}(\mathbb{E}[\mathbf{s}_i \widehat{\mathbf{y}}(\mathbf{x}_i, \boldsymbol{\theta})^T | \mathbf{x}_i, s_i]) - \nabla_{\boldsymbol{\theta}} \text{vec}(\mathbb{E}[\mathbf{s}_i \widehat{\mathbf{y}}(\mathbf{x}_i, \boldsymbol{\theta}')^T | \mathbf{x}_i, s_i])] \right.$$

$$\left. \cdot \text{vec}(W^T \widehat{P}_s^{-1/2}) \right\| \right]$$

$$\leq \beta_\ell \|\boldsymbol{\theta} - \boldsymbol{\theta}'\| + \lambda D^2 \sup_{\mathbf{x}} \sum_{l=1}^m \|\nabla \mathcal{F}_l(\boldsymbol{\theta}, \mathbf{x}) - \nabla \mathcal{F}_l(\boldsymbol{\theta}', \mathbf{x})\|$$

$$+ 2\lambda \sup_{\mathbf{x}, r \in [k]} \left\| \sum_{j=1}^m \nabla \mathcal{F}_j(\boldsymbol{\theta}, \mathbf{x}) - \nabla \mathcal{F}_j(\boldsymbol{\theta}', \mathbf{x}) \hat{p}_S(r)^{-1/2} W_{r,j} \right\|$$

$$\leq \left[ \beta_\ell + 2\lambda \left( D^2 b + \frac{Db}{\sqrt{\hat{p}_S^{\min}}} \right) \right] \|\boldsymbol{\theta} - \boldsymbol{\theta}'\|,$$

by Assumption 1. Combining the above inequalities yields part 1.
2. We have $\nabla_{ww}^2 \widehat{F}(\boldsymbol{\theta}, W) = -2\lambda \widehat{P}_{\hat{y}}$, which is a diagonal matrix with $(\nabla_{ww}^2 \widehat{F}(\boldsymbol{\theta}, W))_{j,j} =$

$-2\lambda\frac{1}{N}\sum_{i=1}^{N}\mathcal{F}_j(\mathbf{x}_i,\boldsymbol{\theta})\leq -2\lambda\hat{p}_{\widehat{Y}}^{\min}$, by Assumption 1. Thus, $\widehat{F}(\cdot,\boldsymbol{\theta})$ is $2\lambda\hat{p}_{\widehat{Y}}^{\min}$-strongly concave for all $\boldsymbol{\theta}$.

3. Our choice of $D$ ensures that $W^*(\boldsymbol{\theta}^*)\in\text{int}(\mathcal{W})$, since

$$\|W^*(\boldsymbol{\theta}^*)\|_F = \|\widehat{P}_s^{-1/2}\widehat{P}_{\hat{y},s}(\boldsymbol{\theta}^*)^T\widehat{P}_{\hat{y}}(\boldsymbol{\theta}^*)^{-1}\|_F \tag{39}$$

$$\leq \frac{1}{\hat{p}_{\widehat{Y}}^{\min}\sqrt{\hat{p}_S^{\min}}}. \tag{40}$$

Therefore, $\max_{W\in\mathcal{W}}\widehat{F}(\boldsymbol{\theta},W) = \max_W\widehat{F}(\boldsymbol{\theta},W)$, which implies part 3 of the lemma.

$\square$

By Assumption 1 and Lemma 11, our choice of $\mathcal{W}$ implies that $W^*(\boldsymbol{\theta}^*)\in\mathcal{W}$ and hence that the solution of Eq. (FERMI obj.) solves

$$\min_{\boldsymbol{\theta}}\max_{W\in\mathcal{W}}\left\{\widehat{F}(\boldsymbol{\theta},W) := \frac{1}{N}\sum_{i\in[N]}\widehat{\mathcal{L}}(\boldsymbol{\theta}) + \lambda\widehat{\Psi}(\boldsymbol{\theta},W)\right\}.$$

This enables us to establish the convergence of Algorithm 1 (which involves projection) to a stationary point for the *unconstrained* min-max optimization problem Eq. (1) that we consider. The $W^t$ projection step in Algorithm 1 is necessary to ensure that the iterates $W^t$ remain bounded, and hence that the smoothness and bounded variance conditions of $\widehat{F}$ are satisfied at every iteration.

### D.1 Proof of Theorem 2

Now we turn to the proof of Theorem 2. We first re-state and prove Proposition 2.

**Proposition 4** (Restatement of Proposition 2). *Let* $\{z_i\}_{i=1}^n = \{\mathbf{x}_i, s_i, y_i\}_{i=1}^n$ *be drawn i.i.d. from an unknown joint distribution* $\mathcal{D}$. *Denote* $\widehat{\psi}_i^{(n)}(\boldsymbol{\theta},W) = -\text{Tr}(W\mathbb{E}[\widehat{\mathbf{y}}(\mathbf{x}_i,\boldsymbol{\theta})\widehat{\mathbf{y}}(\mathbf{x}_i,\boldsymbol{\theta})^T|\mathbf{x}_i]W^T) + 2\text{Tr}\left(W\mathbb{E}[\widehat{\mathbf{y}}(\mathbf{x}_i;\boldsymbol{\theta})\mathbf{s}_i^T|\mathbf{x}_i,s_i]\left(\widehat{P}_s^{(n)}\right)^{-1/2}\right) - 1$, *where* $\widehat{P}_s^{(n)} = \frac{1}{n}\sum_{i=1}^n\text{diag}(\mathbb{1}_{\{s_i=1\}},\cdots,\mathbb{1}_{\{s_i=k\}})$. *Denote* $\Psi(\boldsymbol{\theta},W) = -\text{Tr}(WP_{\hat{y}}W^T) + 2\text{Tr}(WP_{\hat{y},s}P_s^{-1/2}) - 1$, *where* $P_{\hat{y}} = \text{diag}(\mathbb{E}\mathcal{F}_1(\boldsymbol{\theta},\mathbf{x}),\cdots,\mathbb{E}\mathcal{F}_m(\boldsymbol{\theta},\mathbf{x}))$, $(P_{\hat{y},s})_{j,r} = \mathbb{E}_{\mathbf{x}_i,s_i}[\mathcal{F}_j(\boldsymbol{\theta},\mathbf{x}_i)\mathbf{s}_{i,r}]$ *for* $j\in[m], r\in[k]$, *and* $P_s = \text{diag}(P_S(1),\cdots,P_S(k))$. *Assume* $p_S(r) > 0$ *for all* $r\in[k]$. *Then,*

$$\max_W\Psi(\boldsymbol{\theta},W) = D_R(\widehat{Y}(\boldsymbol{\theta});S)$$

*and*

$$\lim_{n\to\infty}\mathbb{E}[\widehat{\psi}_i^{(n)}(\boldsymbol{\theta},W)] = \Psi(\boldsymbol{\theta},W).$$

*Proof.* The first claim, that $\max_W\Psi(\boldsymbol{\theta},W) = D_R(\widehat{Y}(\boldsymbol{\theta});S)$ is immediate from Proposition 1 and its proof, by replacing the empirical probabilities with $\mathcal{D}$-probabilities everywhere. For the second claim, we clearly have

$$\mathbb{E}[\widehat{\psi}_i^{(n)}(\boldsymbol{\theta},W)] = \mathbb{E}\left[-\text{Tr}(W\mathbb{E}[\widehat{\mathbf{y}}(\mathbf{x}_i,\boldsymbol{\theta})\widehat{\mathbf{y}}(\mathbf{x}_i,\boldsymbol{\theta})^T|\mathbf{x}_i]W^T)\right] + 2\mathbb{E}\left[\text{Tr}\left(W\mathbb{E}[\widehat{\mathbf{y}}(\mathbf{x}_i;\boldsymbol{\theta})\mathbf{s}_i^T|\mathbf{x}_i,s_i]\left(\widehat{P}_s^{(n)}\right)^{-1/2}\right)\right] - 1 \tag{41}$$

$$= -\text{Tr}(WP_{\hat{y}}W^T) + 2\mathbb{E}\left[\text{Tr}\left(W\mathbb{E}[\widehat{\mathbf{y}}(\mathbf{x}_i;\boldsymbol{\theta})\mathbf{s}_i^T|\mathbf{x}_i,s_i]\left(\widehat{P}_s^{(n)}\right)^{-1/2}\right)\right] - 1,$$

for any $n\geq 1$. Now, $\widehat{P}_s^{(n)}(r)$ converges almost surely (and in probability) to $p_S(r)$ by the strong law of large numbers, and $\mathbb{E}[\widehat{P}_s^{(n)}(r)] = p_S(r)$. Thus, $\widehat{P}_s^{(n)}(r)$ is a consistent estimator of $p_S(r)$. Then by the continuous mapping theorem and the assumption that $p_S(r)\geq C$ for some $C > 0$, we have that $\left(\widehat{P}_s^{(n)}(r)\right)^{-1/2}$ converges almost surely (and in probability) to $p_S(r)^{-1/2}$. Moreover, we claim that there exists $N^*\in\mathbb{N}$ such that for any $n\geq N^*$, $\text{Var}\left(\left(\widehat{P}_s^{(n)}(r)\right)^{-1/2}\right)\leq\frac{2}{C} < \infty$. To see why

this claim holds, note that the definition of almost sure convergence of $\widehat{P}_s^{(n)}(r)$ to $p_S(r)$ implies that, with probability 1, there exists $N^*$ such that for all $n \geq N^*$,

$$\min_{r \in [k]} \widehat{P}_s^{(n)}(r) \geq \min_{r \in [k]} p_S(r) - C/2 \geq C/2.$$

Thus, $\mathrm{Var}\left( \left(\widehat{P}_s^{(n)}(r)\right)^{-1/2} \right) \leq \mathbb{E}\left[ \left(\widehat{P}_s^{(n)}(r)\right)^{-1} \right] \leq \frac{2}{C}$. Therefore $\left(\widehat{P}_s^{(n)}(r)\right)^{-1/2}$ is a consistent estimator with uniformly bounded variance, hence it is asymptotically unbiased: $\lim_{n \to \infty} \mathbb{E}\left[ \left(\widehat{P}_s^{(n)}(r)\right)^{-1/2} \right] = p_S(r)^{-1/2}$. Furthermore, $\left| \left( \mathbb{E}[\widehat{\mathbf{y}}(\mathbf{x}_i;\boldsymbol{\theta})\mathbf{s}_i^T|\mathbf{x}_i, s_i] \left(\widehat{P}_s^{(n)}\right)^{-1/2} \right)_{j,r} \right| \leq$ $\frac{2}{C}$ for all $n \geq N^*$ and $\left( \mathbb{E}[\widehat{\mathbf{y}}(\mathbf{x}_i;\boldsymbol{\theta})\mathbf{s}_i^T|\mathbf{x}_i, s_i] \left(\widehat{P}_s^{(n)}\right)^{-1/2} \right)_{j,r}$ converges almost surely to $\left( \mathbb{E}[\widehat{\mathbf{y}}(\mathbf{x}_i;\boldsymbol{\theta})\mathbf{s}_i^T|\mathbf{x}_i, s_i] (P_S)^{-1/2} \right)_{j,r}$ as $n \to \infty$ (for any $j \in [m], r \in [k]$). Thus, by Lebesgue's dominated convergence theorem, we have

$$\lim_{n \to \infty} \left( \mathbb{E}\left[ \mathbb{E}[\widehat{\mathbf{y}}(\mathbf{x}_i;\boldsymbol{\theta})\mathbf{s}_i^T|\mathbf{x}_i, s_i] \left(\widehat{P}_s^{(n)}\right)^{-1/2} \right] \right)_{j,r} = \mathbb{E}\left[ \mathbb{E}[\widehat{\mathbf{y}}(\mathbf{x}_i;\boldsymbol{\theta})\mathbf{s}_i^T|\mathbf{x}_i, s_i] \lim_{n \to \infty} \left(\widehat{P}_s^{(n)}(r)\right)^{-1/2} \right]$$

$$= \mathbb{E}\left[\mathbf{s}_{i,r}\mathcal{F}_j(\boldsymbol{\theta},\mathbf{x}_i)\right] \lim_{n \to \infty} \left(\widehat{P}_s^{(n)}(r)\right)^{-1/2}$$

$$= (P_{\hat{y},s})_{j,r} \, p_S(r)^{-1/2}$$

$$= (P_{\hat{y},s} P_s^{-1/2})_{j,r}, \tag{42}$$

for all $j \in [m], r \in [k]$. Combining Eq. (41) with Eq. (42) (and using linearity of trace and matrix multiplication) proves the second claim. $\qquad\square$

We are now ready to prove Theorem 2.

*Proof of Theorem 2.* Denote $\Phi(\boldsymbol{\theta}) := \max_W F(\boldsymbol{\theta}, W)$ for the population-level objective $F(\boldsymbol{\theta}, W) := \mathcal{L}(\boldsymbol{\theta}) + \lambda\Psi(\boldsymbol{\theta}, W)$ (using the notation in Proposition 2). Let $\boldsymbol{\theta}^*$ denote the output of the one-pass/sample-without-replacement version of Algorithm 1, run on the modified empirical objective where $\left(\widehat{P}_s^{(n)}\right)^{-1/2}$ is replaced by the true sensitive attribute matrix $P_S^{-1/2}$. That is, $\boldsymbol{\theta}^* \sim \mathbf{Unif}(\boldsymbol{\theta}_1^*, \ldots, \boldsymbol{\theta}_T^*)$, where $\boldsymbol{\theta}_t^*$ denotes the $t$-th iterate of the modified FERMI algorithm just described. Then, given i.i.d. samples, the stochastic gradients are unbiased (with respect to the population distribution $\mathcal{D}$) for any minibatch size, by Corollary 1 and its proof. Further, the without-replacement sampling strategy ensures that the stochastic gradients are independent across iterations. Additionally, the proof of Proposition 2 showed that $\left(\widehat{P}_s^{(n)}\right)^{-1/2}$ converges almost surely to $P_S^{-1/2}$. Thus, there exists $N$ such that if $n \geq N \geq T = \Omega(\epsilon^{-5})$, then $\min_{r \in [k]} \widehat{P}_s^{(n)}(r) > 0$ (by almost sure convergence of $\widehat{P}_s$, see proof of Proposition 2), and

$$\mathbb{E}\|\nabla\Phi(\boldsymbol{\theta}^*)\|^2 \leq \frac{\epsilon}{4}, \tag{43}$$

by Theorem 1 and its proof. Let $\widehat{\boldsymbol{\theta}}_t^{(n)}$ denote the $t$-th iteration of the one-pass version of Algorithm 1 run on the empirical objective (with $\left(\widehat{P}_s^{(n)}\right)^{-1/2}$). Now,

$$\nabla_{\boldsymbol{\theta}}\widehat{\psi}_i(\boldsymbol{\theta}, W) = -\nabla_{\boldsymbol{\theta}}\mathrm{vec}(\mathbb{E}[\widehat{\mathbf{y}}(\mathbf{x}_i, \boldsymbol{\theta})\widehat{\mathbf{y}}(\mathbf{x}_i, \boldsymbol{\theta})^T|\mathbf{x}_i])^T \, \mathrm{vec}(W^T W)$$

$$+ 2\nabla_{\boldsymbol{\theta}}\mathrm{vec}(\mathbb{E}[\mathbf{s}_i\widehat{\mathbf{y}}(\mathbf{x}_i, \boldsymbol{\theta})^T|\mathbf{x}_i, s_i]) \, \mathrm{vec}\left( W^T \left(\widehat{P}_s^{(n)}\right)^{-1/2} \right),$$

which shows that $\widehat{\boldsymbol{\theta}}_t^{(n)}$ is a continuous (indeed, linear) function of $\left(\widehat{P}_s^{(n)}\right)^{-1/2}$ for every $t$. Thus, the continuous mapping theorem implies that $\widehat{\boldsymbol{\theta}}_t^{(n)}$ converges almost surely to $\boldsymbol{\theta}_t^*$ as $n \to \infty$ for every $t \in [T]$. Hence, if $\widehat{\boldsymbol{\theta}}^{(n)} \sim \mathbf{Unif}\left(\widehat{\boldsymbol{\theta}}_1^{(n)}, \ldots, \widehat{\boldsymbol{\theta}}_T^{(n)}\right)$, then $\widehat{\boldsymbol{\theta}}^{(n)}$ converges almost surely to $\boldsymbol{\theta}^*$. Now,

for any $\boldsymbol{\theta}$, let us denote $W(\boldsymbol{\theta}) = \arg\max_W F(\boldsymbol{\theta}, W)$. Recall that by Danskin's theorem Danskin [1966], we have $\nabla\Phi(\boldsymbol{\theta}) = \nabla_{\boldsymbol{\theta}} F(\boldsymbol{\theta}, W(\boldsymbol{\theta}))$. Then,

$$\|\nabla\Phi(\widehat{\boldsymbol{\theta}}^{(n)}) - \nabla\Phi(\boldsymbol{\theta}^*)\|^2 \leq 2\left\|\nabla_\theta F(\widehat{\boldsymbol{\theta}}^{(n)}, W(\widehat{\boldsymbol{\theta}}^{(n)})) - \nabla_\theta F(\boldsymbol{\theta}^*, W(\widehat{\boldsymbol{\theta}}^{(n)}))\right\|^2$$
$$+ 2\left\|\nabla_\theta F(\boldsymbol{\theta}^*, W(\widehat{\boldsymbol{\theta}}^{(n)})) - \nabla_\theta F(\boldsymbol{\theta}^*, W(\boldsymbol{\theta}^*))\right\|^2$$
$$\leq 2\left[\beta^2\|\widehat{\boldsymbol{\theta}}^{(n)} - \boldsymbol{\theta}^*\|^2 + \beta^2\|W(\widehat{\boldsymbol{\theta}}^{(n)}) - W(\boldsymbol{\theta}^*)\|^2\right]$$
$$\leq 2\left[\beta^2\|\widehat{\boldsymbol{\theta}}^{(n)} - \boldsymbol{\theta}^*\|^2 + \frac{2\beta^2 L^2}{\mu^2}\|\widehat{\boldsymbol{\theta}}^{(n)} - \boldsymbol{\theta}^*\|^2\right],$$

where $L$ denotes the Lipschitz parameter of $F$, $\beta$ is the Lipschitz parameter of $\nabla F$, and $\mu$ is the strong concavity parameter of $F(\boldsymbol{\theta}, \cdot)$: see Lemma 11 and its proof (in Appendix D) for the explicit $\beta$, $L$, and $\mu$. We used Danskin's theorem and Young's inequality in the first line, $\beta$-Lipschitz continuity of $\nabla F$ in the second line, and $\frac{2L}{\mu}$-Lipschitz continuity of the $\arg\max_W F(\boldsymbol{\theta}, W)$ function for $\mu$-strongly concave and $L$-Lipschitz $F(\boldsymbol{\theta}, \cdot)$ (see e.g. [Lowy and Razaviyayn, 2021, Lemma B.2]). Letting $n \to \infty$ makes $\|\widehat{\boldsymbol{\theta}}^{(n)} - \boldsymbol{\theta}^*\|^2 \to 0$ almost surely, and hence $\|\nabla\Phi(\widehat{\boldsymbol{\theta}}^{(n)}) - \nabla\Phi(\theta^*)\|^2 \to 0$ almost surely. Furthermore, Danskin's theorem and Lipschitz continuity of $\nabla_\theta F$ implies that $\|\nabla\Phi(\widehat{\boldsymbol{\theta}}^{(n)}) - \nabla\Phi(\boldsymbol{\theta}^*)\|^2 \leq C$ almost surely for some absolute constant $C > 0$ and all $n$ sufficiently large. Therefore, we may apply Lebesgue's dominated convergence theorem to get $\lim_{n\to\infty} \mathbb{E}\|\nabla\Phi(\widehat{\boldsymbol{\theta}}^{(n)}) - \nabla\Phi(\boldsymbol{\theta}^*)\|^2 = \mathbb{E}\left[\lim_{n\to\infty}\|\nabla\Phi(\widehat{\boldsymbol{\theta}}^{(n)}) - \nabla\Phi(\boldsymbol{\theta}^*)\|^2\right] = 0$. In particular, there exists $N$ such that $n \geq N \implies \mathbb{E}\|\nabla\Phi(\widehat{\boldsymbol{\theta}}^{(n)}) - \nabla\Phi(\boldsymbol{\theta}^*)\|^2 \leq \frac{\epsilon}{4}$. Combining this with Eq. (43) and Young's inequality completes the proof. $\qquad\square$

# E  Experiment Details and Additional Results

## E.1  Model description

For all the experiments, the model's output is of the form $O = \mathrm{softmax}(Wx + b)$. The model outputs are treated as conditional probabilities $\mathbf{p}(\hat{y} = i|x) = O_i$ which are then used to estimate the ERMI regularizer. We encode the true class label $Y$ and sensitive attribute $S$ using one-hot encoding. We define $\ell(\cdot)$ as the cross-entropy measure between the one-hot encoded class label $Y$ and the predicted output vector $O$.

We use logistic regression as the base classification model for all experiments in Fig. 1. The choice of logistic regression is due to the fact that all of the existing approaches demonstrated in Fig. 1, use the same classification model. The model parameters are estimated using the algorithm described in Algorithm 1. The trade-off curves for FERMI are generated by sweeping across different values for $\lambda \in [0, 10000]$. The learning rates $\eta_\theta, \eta_w$ is constant during the optimization process and is chosen from the interval $[0.0005, 0.01]$ for all datasets. Moreover, the number of iterations $T$ for experiments in Fig. 1 is fixed to 2000. Since the training and test data for the Adult dataset are separated and fixed, we do not consider confidence intervals for the test accuracy. We generate ten distinct train/test sets for each one of the German and COMPAS datasets by randomly sampling $80\%$ of data points as the training data and the rest $20\%$ as the test data. For a given method in Fig. 1, the corresponding curve is generated by taking the average test accuracy on 10 training/test datasets. Furthermore, the confidence intervals are estimated based on the test accuracy's standard deviation on these 10 datasets.

To perform the experiments in Sec. 3.3 we use a a linear model with softmax activation. The model parameters are estimated using the algorithm described in Sec. 3. The data set is cleaned and processed as described in Kearns et al. [2018]. The trade-off curves for FERMI are generated by sweeping across different values for $\lambda$ in $[0, 100]$ interval, learning rate $\eta$ in $[0.0005, 0.01]$, and number of iterations $T$ in $[50, 200]$. The data set is cleaned and processed as described in Kearns et al. [2018].

For the experiments in Sec. 3.4, we create the synthetic color MNIST as described by Li and Vasconcelos [2019]. We set the value $\sigma = 0$. In Fig. 6, we compare the performance of stochastic solver (Algorithm 1) against the baselines. We use a mini-batch of size 512 when using the stochastic solver. The color MNIST data has 60000 training samples, so using the stochastic solver gives a speedup of around 100x for each iteration, and an overall speedup of around 40x. We present our results on two neural network architectures; namely, LeNet-5 Lecun et al. [1998] and a Multi-layer perceptron (MLP). We set the MLP with two hidden layers (with 300 and 100 nodes) and an output layer with ten nodes. A ReLU activation follows each hidden layer, and a softmax activation follows the output layer.

Some general advice for tuning $\lambda$: Larger value for $\lambda$ generally translates to better fairness, but one must be careful to not use a very large value for $\lambda$ as it could lead to poor generalization performance of the model. The optimal values for $\lambda$, $\eta$, and $T$ largely depend on the data and intended application. We recommend starting with $\lambda \approx 10$. In Appendix E.4, we can observe the effect of changing $\lambda$ on the model accuracy and fairness for the COMPAS dataset.

## E.2  More comparison to Mary et al. [2019]

The algorithm proposed by Mary et al. [2019] backpropagates the batch estimate of ERMI, which is biased especially for small minibatches. Our work uses a correct and unbiased implementation of a stochastic ERMI estimator. Furthermore, *Mary et al. [2019] does not establish any convergence guarantees, and in fact their algorithm does not converge*. See Fig. 7 for the evolution of *training loss* (i.e. value of the objective function) and *test accuracy*. For this experiment, we follow the same setup used in [Mary et al., 2019, Table 1]; the minibatch size for this experiment is 128.

## E.3  Performance in the presence of outliers & class-imbalance

We also performed an additional experiment on Adult (setup of Fig 1) with a random $10\%$ of sensitive attributes in *training* forced to 0. FERMI offers the most favorable tradeoffs on *clean test* data, however, all methods reach a higher plateau (see Fig 8). The interplay between fairness, robustness,

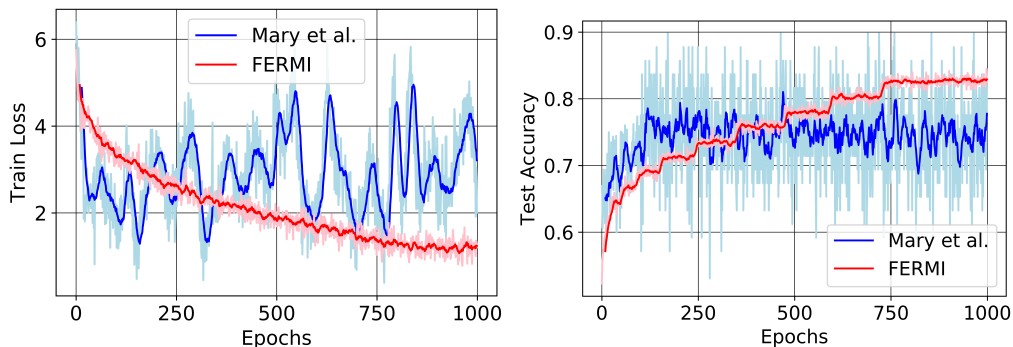

Figure 7: Mary et al. [2019] fails to converge to a stationary point whereas our stochastic algorithm easily converges.

and generalization is an important future direction. With respect to imbalanced sensitive groups, the experiments in Fig 5 are on a naturally imbalanced dataset, where $\max_{s \in \mathcal{S}} p(s) / \min_{s \in \mathcal{S}} p(s) > 100$ for 3-18 sensitive attrib, and FERMI offers the favorable tradeoffs.

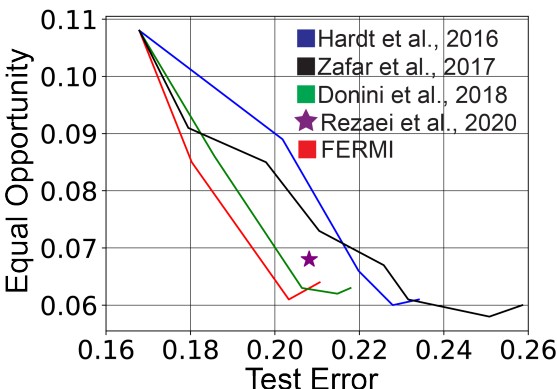

Figure 8: Comparing FERMI with other methods in the presence of outliers (random 10% of sensitive attributes in *training* forced to 0. FERMI still achieves a better trade-off compared to all other baselines.

### E.4    Effect of hyperparameter $\lambda$ on the accuracy-fairness tradeoffs

We run ERMI algorithm for the binary case to COMPAS dataset to investigate the effect of hyperparameter tuning on the accuracy-fairness trade-off of the algorithm. As it can be observed in Fig. 9, by increasing $\lambda$ from 0 to 1000, test error (left axis, red curves) is slightly increased. On the other hand, the fairness violation (right axis, green curves) is decreased as we increase $\lambda$ to 1000. Moreover, for both notions of fairness (demographic parity with the solid curves and equality of opportunity with the dashed curves) the trade-off between test error and fairness follows the similar pattern. To measure the fairness violation, we use demographic parity violation and equality of opportunity violation defined in Section equation 3 for the solid and dashed curves respectively.

### E.5    Complete version of Figure 1 (with pre-processing and post-processing baselines)

In Figure 1 we compared FERMI with several state-of-the-art in-processing approaches. In the next three following figures we compare the in-processing approaches depicted in Figure 1 with pre-processing and post-processing methods including Hardt et al. [2016], Kamiran et al. [2010], Feldman et al. [2015].

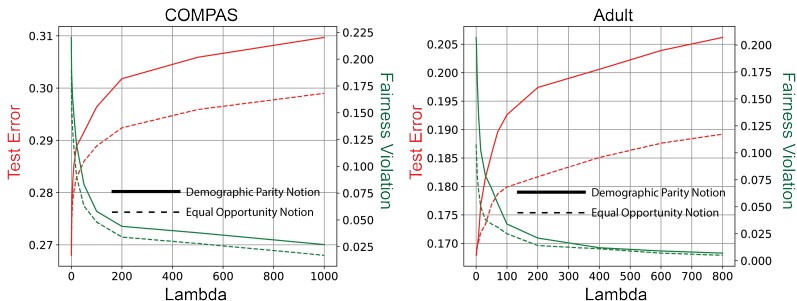

Figure 9: Tradeoff of fairness violation vs test error for FERMI algorithm on COMPAS and Adult datasets. The solid and dashed curves correspond to FERMI algorithm under the demographic parity and equality of opportunity notions accordingly. The left axis demonstrates the effect of changing $\lambda$ on the test error (red curves), while the right axis shows how the fairness of the model (measured by equality of opportunity or demographic parity violations) depends on changing $\lambda$.

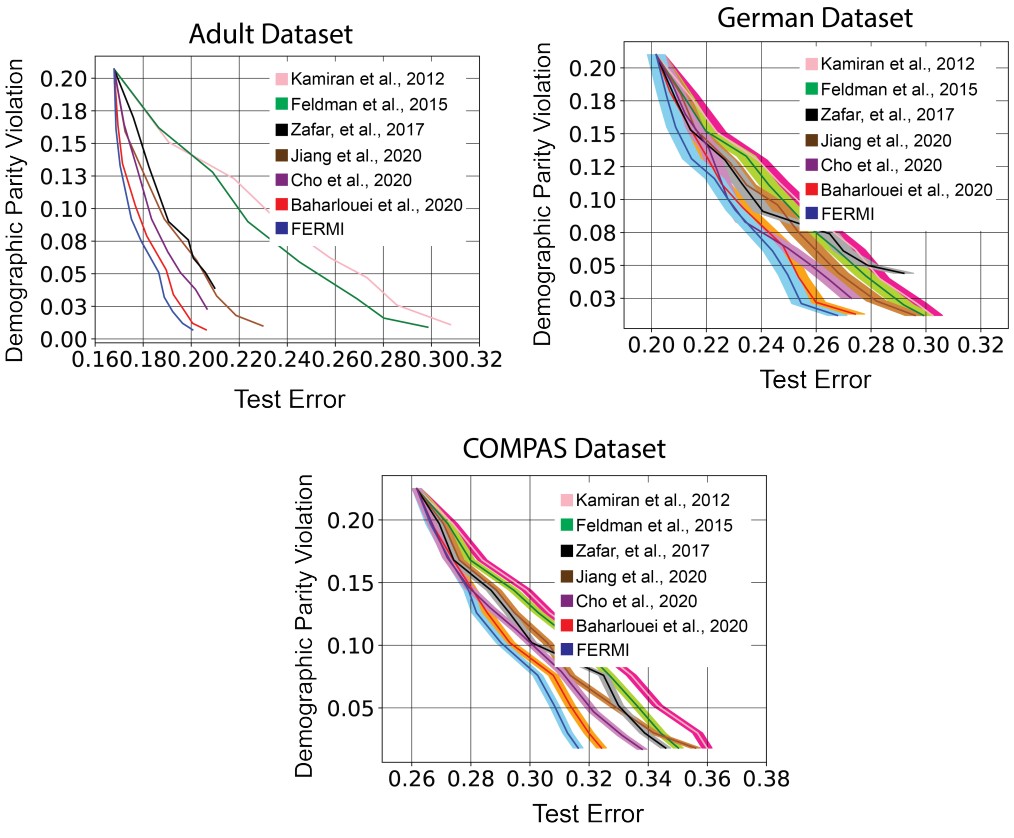

Figure 10: Tradeoff of demographic parity violation vs test error for FERMI algorithm on COMPAS, German, and Adult datasets.

### E.6 Description of datasets

All of the following datasets are publicly available at UCI repository.

**German Credit Dataset.**[11] German Credit dataset consists of 20 features (13 categorical and 7 numerical) regarding to social, and economic status of 1000 customers. The assigned task is to classify customers as good or bad credit risks. Without imposing fairness, the DP violation of the

---

[11]https://archive.ics.uci.edu/ml/datasets/statlog+(german+credit+data)

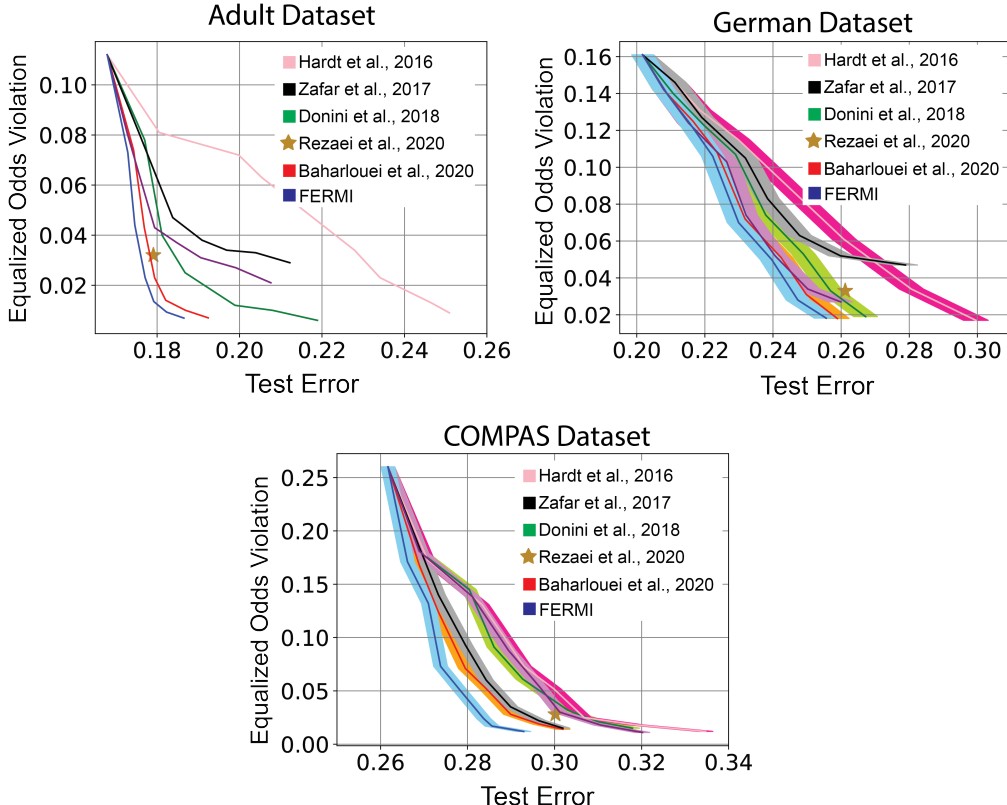

Figure 11: Tradeoff of equalized odds violation vs test error for FERMI algorithm on COMPAS, German, and Adult datasets.

trained model is larger than $20\%$. We choose $80\%$ of customers as the train data and the remaining $20\%$ customers as the test data. The sensitive attributes are gender, and marital-status.

**Adult Dataset.**[12] Adult dataset contains the census information of individuals including education, gender, and capital gain. The assigned classification task is to predict whether a person earns over 50k annually. The train and test sets are two separated files consisting of $32,000$ and $16,000$ samples respectively. We consider gender and race as the sensitive attributes (For the experiments involving one sensitive attribute, we have chosen gender). Learning a logistic regression model on the training dataset (without imposing fairness) shows that only 3 features out of 14 have larger weights than the gender attribute. Note that removing the sensitive attribute (gender), and retraining the model does not eliminate the bias of the classifier. the optimal logistic regression classifier in this case is still highly biased. For the clustering task, we have chosen 5 continuous features (Capital-gain, age, fnlwgt, capital-loss, hours-per-week), and $10,000$ samples to cluster. The sensitive attribute of each individual is gender.

**Communities and Crime Dataset**.[13] The dataset is cleaned and processed as described in Kearns et al. [2018]. Briefly, each record in this dataset summarizes aggregate socioeconomic information about both the citizens and police force in a particular U.S. community, and the problem is to predict whether the community has a high rate of violent crime.

**COMPAS Dataset**.[14] Correctional Offender Management Profiling for Alternative Sanctions (COMPAS) is a famous algorithm which is widely used by judges for the estimation of likelihood of reoffending crimes. It is observed that the algorithm is highly biased against the black defendants.

---

[12] https://archive.ics.uci.edu/ml/datasets/adult.

[13] http://archive.ics.uci.edu/ml/datasets/communities+and+crime

[14] https://www.kaggle.com/danofer/compass

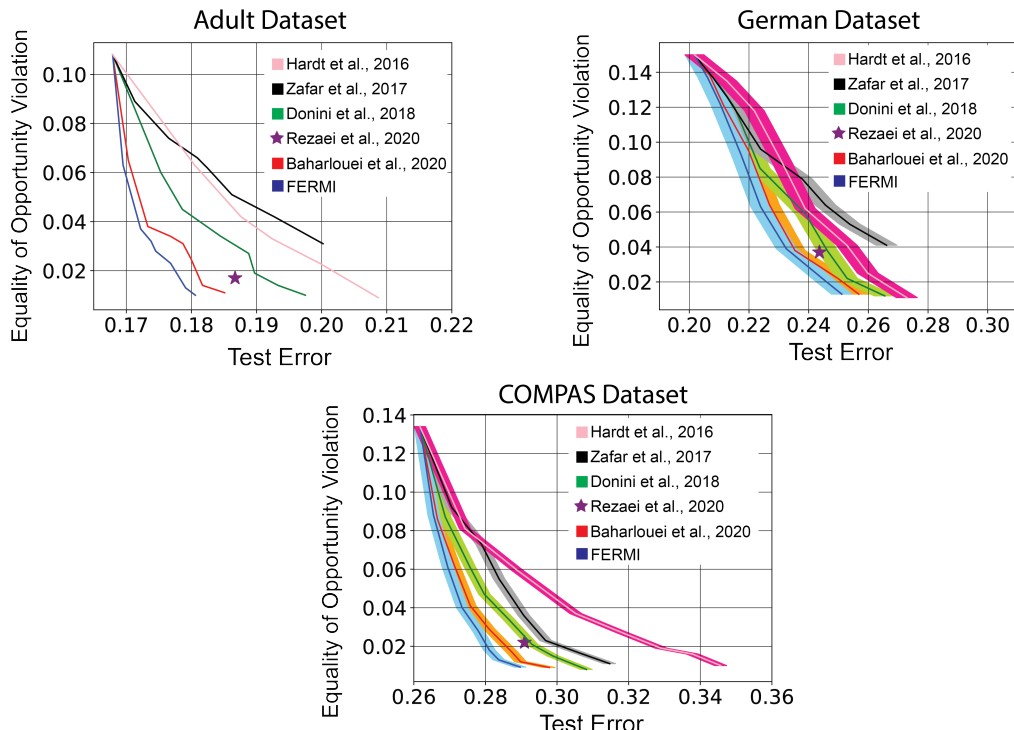

Figure 12: Tradeoff of equality of opportunity violation vs test error for FERMI algorithm on COMPAS, German, and Adult datasets.

The dataset contains features used by COMPAS algorithm alongside with the assigned score by the algorithm within two years of the decision.

**Colored MNIST Dataset.**[15] We use the code by Li and Vasconcelos [2019] to create a Colored MNIST dataset with $\sigma = 0$. We use the provided LeNet-5 model trained on the colored dataset for all baseline models of Baharlouei et al. [2020], Mary et al. [2019], Cho et al. [2020a] and FERMI, where we further apply the corresponding regularizer in the training process.

---

[15]https://github.com/JerryYLi/Dataset-REPAIR/

# F   Code for Experiments

The code for all of the experiments in this paper is available on Dropbox:
https://www.dropbox.com/sh/516cm8olq0idpsd/AADD0LOcPWpx4AAhzsEkFTOca?dl=0

