# OpenReview forum: "A Stochastic Optimization Framework for Fair Risk Minimization"
_NeurIPS.cc/2022/Workshop/TSRML — TSRML2022_

### Official Review · Reviewer_zcvD · 2022-10-11

**Overall Rating:** 8

**Summary:**

This paper studies the problem of fair empirical risk minimization for large-scale ML problems. The authors minimize a loss function regularized with exponential Rényi mutual information and show that their methods converge to a stationary point, making their method the first stochastic in-processing fairness algorithm with guaranteed convergence. Their procedure relies on deriving an unbiased stochastic gradient estimators for their objective and using stochastic optimization to minimize their objective. In addition to theoretical results, the authors empirically evaluate the performance of their algorithm for different notions of fairness on tasks such as binary and non-binary classification.

**Strengths:**

The main contribution of this paper is the performance guarantees of their algorithm, as they are the first to provide a stochastic in-processing fairness algorithm with guaranteed convergence. Additionally, the experiments complement the theory nicely and the paper is well-written.

**Weaknesses:**

I see no major weaknesses.

**Overall Recommendation:**

Overall, this is a good paper which is very relevant to the workshop. I recommend acceptance.

**Review Confidence:**

3: The reviewer is fairly confident that the evaluation is correct

---

### Official Review · Reviewer_hqQZ · 2022-10-20
**A notable work which incorporate fairness constraints to stochastic optimization problem**

**Overall Rating:** 6

**Summary:**

They proposed the first stochastic in-processing fairness algorithm with a convergence guarantee. Their theoretical method is validated by experimental analysis which shows the best tradeoff of fairness and accuracy compared to the other baseline. They tried to solve the optimization function containing the fairness constraint using a novel approach





**Strengths:**

1. Fair empirical risk minimization is an understudied topic and this work studies this topic and introduces a method to incorporate fairness into the stochastic optimization function. Where the novelty of this work is deriving a statistically unbiased stochastic estimator to develop a stochastic convergent algorithm solving the stochastic optimization function with fairness constraints.

2. The results surpass other baseline methods in the test dataset and obtained a better fairness-accuracy tradeoff (using different well-known fairness notions). Unlike the previous methods, this method attained a result better than the random method (Naive baseline) even with a very small batch of data.

3. I think this paper has good results from the both worlds of theoretical and experimental analysis.

**Weaknesses:**

Since randomness is coming from the mini-batch sampling, I think it is useful to mention how you overcome this randomness in the main paper. For example, some parts of Appendix G.1 explaining the experiment setting are better to move to the main paper so that reader can follow the experiments and the importance of the results. However, I understand that authors tried to put a lot of information in a limited number of pages.

**Overall Recommendation:**

The work is intriguing to me. Specifically, the extensive experimental analysis which shows a better trade-off of fairness-accuracy compared to the other baseline methods. In sense of theoretical analysis, the paper has a novelty in including the fairness constraint in the stochastic optimization function. Overall, I enjoyed reading this article and I think the authors provided sufficient evidence to confirm their theoretical results.

**Review Confidence:**

2: The reviewer is willing to defend the evaluation, but it is quite likely that the reviewer did not understand central parts of the paper

---

### Official Review · Reviewer_WYhP · 2022-10-21
**Well-written, interesting paper on a stochastic algorithm for balancing fairness and accuracy**

**Overall Rating:** 8

**Summary:**

This paper provides a method for empirical risk minimization (ERM) by proposing a stochastic optimization algorithm for balancing accuracy and fairness. The authors introduce regularization of the discriminatory loss with empirical Renyi mutual information between the predictions and the discrete valued sensitive variables. They show that the proposed algorithm, FERMI, provably converges. They describe how the Renyi mutual information term can be modified for three kinds of fairness notions - demographic parity, equalized odds and equality of opportunity. The framework works for multi-class classification and multiple sensitive attributes with, even under low batch sizes. They compare the fairness-accuracy trade-off of FERMI with other in-processing methods on Adult, German credit and COMPAS datasets.

**Strengths:**

The framework is neat, and works well for non-binary classes and sensitive attributes. In addition, the stochastic algorithm comes with convergence guarantees. The experiments indicate that FERMI possesses a better fairness-accuracy trade-off in comparison to other recent in-processing algorithms. The paper is well-written.

**Weaknesses:**

The experimental evaluation could be performed with more complex datasets. It would be interesting to see how the fairness-accuracy tradeoff of FERMI compares to other SOTA methods such as 1) adversarial debiasing (Zhang et al., 2018) and 2) Alghamdi, W., Hsu, H., Jeong, H., Wang, H., Michalak, P. W., Asoodeh, S., & Calmon, F. P. (2022). Beyond adult and compas: Fairness in multi-class prediction. arXiv preprint arXiv:2206.07801. In particular the work by Alghamdi et al. stated above is an algorithm with convergence guarantees for multi-class classification, and it would be nice to see a comparison, particularly with larger datasets.

Although the authors provide a nice comparison in Table 1, it is worth noting that the proposed method works only for discrete valued sensitive attributes, unlike some other SOTA methods mentioned such as 1) Mary et al. 2019, 2) Vincent Grari, Sylvain Lamprier, Marcin Detyniecki, Fairness-Aware Neural Rényi Minimization for Continuous Features. IJCAI 2020: 2262-2268, which are designed for continuous attributes.


**Overall Recommendation:**

Overall, this is a balanced paper with interesting contributions. I recommend this paper be accepted.

**Review Confidence:**

4: The reviewer is confident but not absolutely certain that the evaluation is correct

---

### Decision · Program_Chairs · 2022-10-23

**Decision:**

Accept

**Comment:**

Solid work on regularization based fair learning with guaranteed convergence.